# FoR-SALE: Frame of Reference-guided Spatial Adjustment in LLM-based Diffusion Editing

## Abstract

Frame of Reference (FoR) is a fundamental concept in spatial reasoning that humans utilize to comprehend and describe space. With the rapid progress in Multimodal Language models, the moment has come to integrate this long-overlooked dimension into these models. In particular, in text-to-image (T2I) generation, even state-of-the-art models exhibit a significant performance gap when spatial descriptions are provided from perspectives other than the camera. To address this limitation, we propose **F**rame **o**f **R**eference-guided **S**patial **A**djustment in **L**LM-based Diffusion **E**diting (FoR-SALE), an extension of the Self-correcting LLM-controlled Diffusion (SLD) framework for T2I. For-Sale evaluates the alignment between a given text and an initially generated image, and refines the image based on the Frame of Reference specified in the spatial expressions. It employs vision modules to extract the spatial configuration of the image, while simultaneously mapping the spatial expression to a corresponding camera perspective. This unified perspective enables direct evaluation of alignment between language and vision. When misalignment is detected, the required editing operations are generated and applied. FoR-SALE applies novel latent-space operations to adjust the facing direction and depth of the generated images. We evaluate FoR-SALE on two benchmarks specifically designed to assess spatial understanding with FoR. Our framework improves the performance of state-of-the-art T2I models by up to 5.3% using only a single round of correction.

## 1 Introduction

Spatial understanding refers to the ability to comprehend the location of objects within a space. This ability is fundamental to human cognition and everyday tasks. A key component of this ability is comprehending the expressed Frame of Reference (FoR) that indicates the perspective from which spatial relations are interpreted. While extensively studied in cognitive linguistics (Mou & McNamara, 2002; Levinson, 2003; Tenbrink, 2011; Coventry et al., 2018), FoRs have received limited attention in AI models, particularly within Multimodal Large Language Models (MLLMs) (Liu et al., 2023a; Chen et al., 2024). Recent studies highlight substantial shortcomings in reasoning over FoR by MLLMs across multiple tasks, such as Visual Question Answering (Zhang et al., 2025b), Text-to-Image (T2I) generation (Wang et al., 2025b), and text-based

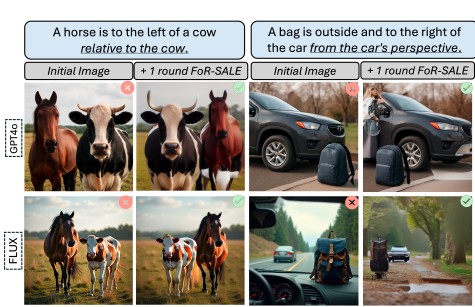

Figure 1: Examples of images generated by SOTA T2I models and the corresponding outputs after one round of correction using FoR-SALE.

QA (Premsri & Kordjamshidi, 2025). One problem domain that highlights the lack of reasoning over FoR is T2I generation in diffusion models. Wang et al. (2025b) and Premsri & Kordjamshidi (2025) show that diffusion models exhibit substantially lower spatial alignment when spatial expressions are described from non-camera perspectives. As illustrated in Figure 1, even SOTA T2I models—GPT-4o (OpenAI, 2025a) and FLUX.1(Black Forest Labs, 2025)—struggle to correctly generate images that reflect spatial relations described from non-camera perspectives. To address

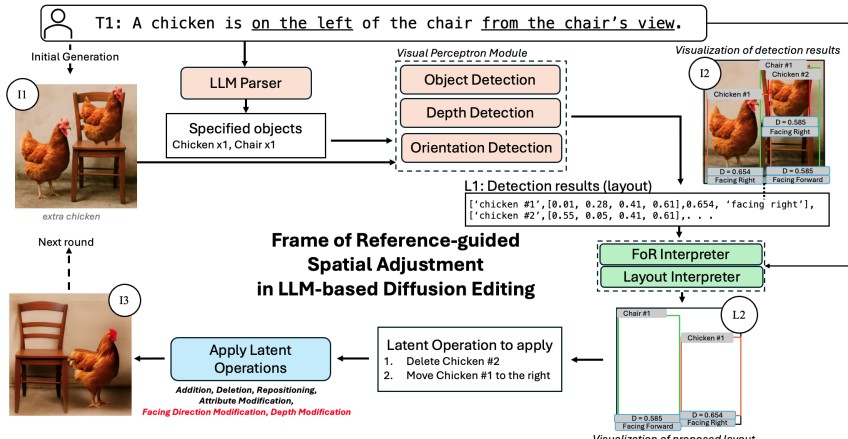

Figure 2: Overview of the FoR-SALE pipeline. It begins by extracting layout information from the initial image using an LLM Parser and a Visual Perception Module. This information is then passed through the FoR-Interpreter and Layout Interpreter to generate a revised layout. A sequence of latent operations is then derived by comparing the initial layout with revised layouts and applied to synthesize an updated image. The resulting image can undergo additional refinement rounds if needed.

this issue, we propose the **F**rame **o**f **R**eference-guided **S**patial **A**djustment in **L**LM-based Diffusion **E**diting (FoR-SALE) framework. Our approach builds upon the Self-correcting LLM-controlled Diffusion (SLD) pipeline (Wu et al., 2024), which uses LLMs to validate prompts and generate suggested layouts for editing images through latent-space operations. However, the original SLD framework does not account for FoR, limiting its ability to handle spatial prompts grounded in perspectives other than the camera view. FoR-SALE extends this paradigm by explicitly modeling FoR and enabling spatial adjustment over diverse perspective conditions.

Figure 2 illustrates the FoR-SALE pipeline. The process begins with standard T2I generation, where a context ($T_1$) is passed to a T2I module to produce an initial image ($I_1$). Meanwhile, the LLM parser extracts the key object from the given text. Then, the key objects are passed to the Visual Perception Module to extract three types of visual properties, namely object location, orientation, and depth. The extracted visual properties ($I_2$) are then converted into a textual format ($L_1$). The input expression ($T_1$) along with textual layout information ($L_1$) is fed to the FoR Interpreter, which first identifies the frame of reference and converts the expression into the camera's perspective—a unified viewpoint. Subsequently, the Layout LLM is employed to generate a suggested layout ($L_2$) in textual form that aligns with the updated spatial expression. Next, the suggested layout is compared with the visual detection outputs ($L_1$) to identify mismatches, which are used to formulate self-correction operations, such as adjusting an object's facing direction or depth. These corrections are applied in the latent space during image synthesis using the Stable Diffusion model. Note that these operations are generic and can be applied to other diffusion models. Finally, a new image is generated from the corrected latent representation, ensuring consistency with the spatial configuration described in the input—particularly for the specified FoR. The resulting image ($I_3$) can undergo additional refinement rounds if needed.

We demonstrate the effectiveness of FoR-SALE using two benchmarks: FoR-LMD, a modification of the LMD (Lian et al., 2024) benchmark that includes perspective, and FoREST (Premsri & Kordjamshidi, 2025), a benchmark that includes textual input for various FoR cases. We observed that our technique can improve images generated from SD-3.5-large, FLUX.1, and GPT-4o, SOTA models of T2I tasks, up to 5.30% improvement in a single correction round and $9.90\%$ in three rounds. Moreover, we provide a thorough analysis to highlight both the limitations of T2I models and LLMs used to suggest layouts from different perspectives. Our contribution[1] can be summarized as follows, **1.** We propose the first self-image correction framework that incorporates the notion of frame of reference (FoR) in T2I generation. **2.** We introduce novel editing operations within a self-correcting

---

[1]Code will be available upon acceptance.

framework to handle various FoRs in generated images. **3.** We augment an existing benchmark to enable evaluation of FoR understanding in T2I models, and conduct a comprehensive evaluation across multiple T2I and self-correction frameworks. Our model achieves SOTA performance when applied to images generated by GPT-4o.

## 2    RELATED WORKS

**Frame of Reference in MLLMs.** Multiple benchmarks have been developed to evaluate the spatial understanding of MLLMs across various tasks (Anderson et al., 2018; Mirzaee et al., 2021; Mirzaee & Kordjamshidi, 2022; Shi et al., 2022; Cho et al., 2023). However, most of these benchmarks overlook the concept of FoR. Only a few recent benchmarks explicitly address FoR-related reasoning (Liu et al., 2023a; Chen et al., 2024; Zhang et al., 2025a; Wang et al., 2025a). For example, Liu et al. (2023a) shows that training a vision-language model with text that includes FoR information can improve visual question answering (VQA). Wang et al. (2025a) introduces a comprehensive benchmark for spatial VQA that incorporates FoR examples, though FoR is not its central focus of evaluation. Three recent studies focus more directly on evaluating FoR understanding in MLLMs. First, Zhang et al. (2025b) assesses FoR handling in VQA settings and reveals substantial limitations, especially when reasoning goes beyond the default camera-centric view. Second, Premsri & Kordjamshidi (2025) investigates FoR reasoning in natural language prompts—both ambiguous and unambiguous—and finds persistent failures in both question answering and layout generation when the perspective diverges from the camera view. Third, Wang et al. (2025b) conducts a comprehensive evaluation of T2I models and finds that even SOTA models fail to preserve correct spatial relations when the context is not grounded in the camera's perspective and includes 3D information such as orientation and distance. In this work, we extend this line of research by providing a new evaluation of T2I models based on their alignment with FoR-grounded spatial expressions. We also enhance the T2I models in comprehending varying FoR conditions.

**Spatial Alignment in T2I.** Several studies have sought to improve the spatial alignment of T2I models with user input. Early approaches introduced predefined spatial constraints—such as depth maps (Zhang et al., 2023; Mo et al., 2024), object layouts (Li et al., 2023), or attention maps (Wang et al., 2024a; Pang et al., 2024)—to guide image generation. However, these often require manual configuration or model retraining to interpret the constraints. With advances in spatial reasoning from LLMs, recent work has leveraged them to generate spatial guidance automatically. For example, Cho et al. (2023) uses an LLM to generate initial layouts that guide diffusion models without additional training. More recent methods employ MLLMs to control 3D spatial arrangements by generating feedback used for reinforcement training of diffusion models (Liu et al., 2025), train a T2I model using compositional questions derived from the input (Sun et al., 2025), or produce action plans for sequential editing (Wu et al., 2024; Goswami et al., 2024). While these methods are promising, they ignore the reasoning issues across FoR variations. In contrast, we explicitly address this by extending the SLD framework (Wu et al., 2024) to support editing under diverse FoRs.

## 3    METHODOLOGY

In this section, we explain our proposed FoR-SALE, an extension of the SLD framework (Wu et al., 2024). An overview of the framework is illustrated in Figure 2. FoR-SALE follows the SLD framework, which consists of two main components: (1) LLM-driven visual perception and (2) LLM-controlled layout interpretation. However, we adapt the two components to accommodate more fine-grained perception and layout interpretation for recognizing FoR and correcting the image accordingly.

### 3.1    LLM-DRIVEN VISUAL PERCEPTION MODULE

The process begins with standard T2I generation, where a textual input is passed to a T2I model to create an image. The FoR-SALE then proceeds by extracting necessary information from both the spatial expression using an LLM parser and the generated image using a visual perception module.

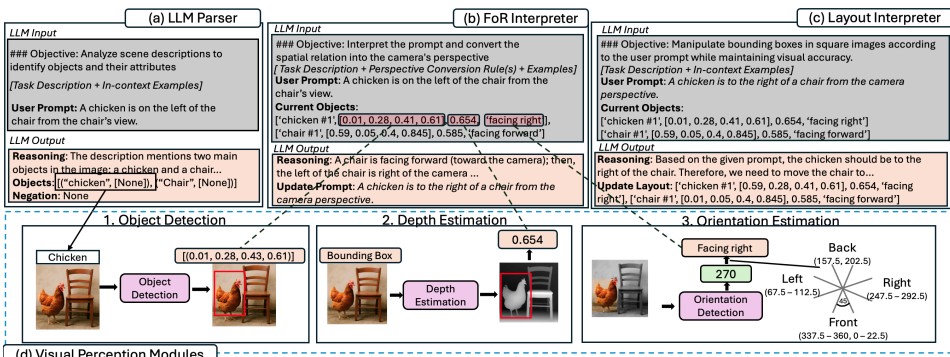

Figure 3: Example inputs and outputs from the LLM Parser, FoR Interpreter, Layout Interpreter, and Visual Perception Module. The LLM Parser output guides the Visual Perception Module in extracting object-specific information, including bounding boxes, orientation, and depth. This information is passed to the FoR Interpreter, which converts the spatial expression to the camera's perspective. The Layout Interpreter then generates a suggested spatial layout based on the updated prompt.

### 3.1.1 LLM PARSER

In this first step, we prompt an LLM to extract a list of key object mentions and their attributes from the input text, denoted as $L$. To facilitate accurate extraction, we provide the LLM with textual instructions and in-context examples. For example, given the spatial expression *A red chicken is on the left of a chair from the chair's view*. The output of LLM is $L$ = *("chicken", ["red"]), ("chair", [None])* where "red" is the attribute associated with the chicken, and "None" indicates that no specific attribute is mentioned for the chair. All prompt specifications are provided in Appendix I.

### 3.1.2 VISUAL PERCEPTION MODULE

The obtained list $L$ is fed into the visual perception module in the SLD framework with an open-vocabulary object detection. In our FoR-SALE, we add new visual perception components to deal with FoR. These include depth estimation and orientation detection. Figure 3 (d) illustrates this module. The open-vocabulary object detector receives information in $L$ with the following prompt format "image of a/an [attribute] [object name]" and outputs bounding boxes, denoted as $B$. The outputs are represented in the following list format, ((attribute) (object name) (#object ID), $[x, y, w, h]$) where $(x, y)$ indicates the coordinates of the upper-left corner of the bounding box from 0.0 to 1.0, $w$ is its width, and $h$ is its height. The object ID is a serial number assigned uniquely to each detected object. Next, the depth estimation model is used to predict the depth map of the image, denoted as $D$. To extract object-specific depth values, denoted as $D_i$, a segmentation mask is applied using the bounding boxes from $B$ and computes the average pixel depth within each masked region using the following equation,$D_i = \sum_j^R d_j/|R|$ where $i$ is id of the object, $R$ is the mask region of the object, and $d_j$ is depth at pixel $j$. The value of $D_i$ ranges from 0 to 1. Finally, an orientation detection model is invoked over the object segmentation to obtain the orientation angle of the object. This angle is then converted into a facing direction, denoted as $f_i$. There are eight facing direction categories: $orientation$={ForwardLeft, Left, BackwardLeft, Back, BackwardRight, Right, ForwardRight, Front}. Each category spans a 45-degree range, starting from 22.5° to 67.5° for ForwardLeft, and continuing in 45° intervals for the remaining orientation labels. We collect these visual information about each object and obtain a new list with these detail in a new format, denoted $V_L = \{($(attribute) (object name) (#object ID)$, [x, y, w, h], D_i, f_i)\}$. An example of representation can be found in Figure 2.

## 3.2 LLM CONTROLLED DIFFUSION

After obtaining visual information ($V_L$), two additional modules are employed to analyze and modify the image, that is, LLM-Interpreters and Image Correction.

### 3.2.1 LLM-INTERPRETERS

This module analyzes $V_L$ together with the input text $T$ and proposes a revised layout, denoted as $\tilde{V}_L$ in the same format. The original SLD framework employs an LLM for layout interpretation. However, in FoR-SALE, we incorporate one additional LLM, that is, FoR interpreter. Figure 3 (b) and (c) illustrate these two LLMs.

**1) FoR-Interpreter.** Based on the findings of Zhang et al. (2025b), Premsri & Kordjamshidi (2025), and Wang et al. (2025b), MLLMs demonstrate significantly stronger performance when reasoning over spatial expressions described from the camera perspective. Motivated by this observation, we hypothesize that converting the perspective of the spatial expressions into a camera viewpoint can alleviate this issue. The input to FoR-Interpreter consists of the spatial text, $T$, and visual information of the generated image, $V_L$. The output is a spatial expression rewritten from the camera perspective, denoted as $T'$. If no spatial relation is present, the model returns the input text unchanged. We provide an in-context information scheme for the FoR-Interpreter to conduct this perspective conversion. In particular, we include spatial perspective conversion rules. A total of 32 rules are manually defined—one for each combination of the eight facing directions considered in the Visual Perception Module and four spatial relations (front, back, left, right). e.g., *if the object is facing left, the left side of the object is in front of the camera*. These rules cover directional spatial relations that are most strongly impacted by FoR interpretation, along with the eight possible facing directions. This set of spatial relations is based on qualitative directional relations, which are a closed set making the formalization feasible Kordjamshidi et al. (2010). All 32 rules are included in the Appendix. An example of the input and output of the FoR-Interpreter is shown in Figure 3(b).

**2) Layout Interpreter.** After obtaining the spatial expression, $T'$, that follows the camera perspective, the second LLM uses $T'$ and $V_L$ as input to analyze the layout. The Layout-Interpreter LLM is prompted with manually crafted in-context examples to analyze whether the current layout aligns with the provided $T'$. If misalignment is detected, the LLM is instructed to propose a revised layout $\tilde{V}_L$ that satisfies the spatial description. An example of the input and output is shown in Figure 3(c).

### 3.2.2 IMAGE CORRECTION

In this step, we compare the current layout $V_L$ with the proposed layout $\tilde{V}_L$ using an exact matching process to detect the misalignment. If there is any misalignment between the two layouts, we create a sequence of editing operations to modify the image and align it with $\tilde{V}_L$. The original SLD framework includes four editing operations: Addition, Deletion, Reposition, and Attribute Modification. Our framework extends this set by introducing **two new operations for handling FoR**, that is, Facing Direction Modification and Depth Modification. Before applying any operation, backward diffusion (Ho et al., 2020) is performed on the initial image to obtain its latent representation, which serves as the basis for all subsequent editing actions. After all editing actions are applied, Stable Diffusion is called to synthesize the final image.

**1) Addition.** Following the prior framework by (Wu et al., 2024), this operation involves two main steps. First, it generates the target object within the designated bounding box area using base Stable Diffusion, and then generates the object's segment using SAM (Kirillov et al., 2023). Next, we perform a backward diffusion process with the base diffusion model over the generated object region to extract a new object latent representation. This object-specific latent representation is then merged into the latent space of the original image to complete the composition.

**2) Deletion.** The process first segments the object using SAM within its bounding box. The latent representation corresponding to the segmented region is then removed and replaced with Gaussian noise. This replacement allows the object's region to be reconstructed during the final diffusion step.

**3) Reposition.** To preserve the object's aspect ratio, this step begins by shifting and resizing the object from its original bounding box to the new target bounding box. After repositioning, SAM is used to do object segmentation. Then, a backward diffusion process is used to obtain the latent representation. This new representation is then integrated into the latent space of the original image at the updated location. To remove the object from the original position, we replace the corresponding latent region, identified via SAM at the original bounding box, with Gaussian noise before the final diffusion step.

**4) Attribute Modification.** To edit an object's attribute, it begins by employing SAM to segment the object region within its bounding box. An attribute modification diffusion model, e.g., DiffEdit (Couairon et al., 2023), is then called with a new prompt to modify the object's attribute within the defined region. For example, calling DiffEdit with the prompt "a red car" modifies the color of a car in the specified region to red. After the attribute is edited, a backward diffusion process is performed to extract the corresponding latent representation. This updated latent is then integrated into the image latent space to complete the modification.

**5) Facing direction Modification.** We introduce this new operation that begins by using SAM to segment the object's region. Then it invokes the DiffEdit with a prompt specifying the desired facing direction to generate an image of the object with the new orientation. Next, the base diffusion model is used to perform a backward diffusion process for obtaining the latent representation of the reoriented object. Finally, this latent is integrated into the overall image latent space to complete the modification.

**6) Depth Modification.** We introduce this new operation that begins by synthesizing the new depth of the given object using the equation, $d_{j'} = min(1, max(0, d_j - D_i + D_{i'}))$, where $d_j, d_{j'}$ denote the original and updated depth values of pixel $j$, respectively. $D_i$ represents the current average depth of object $i$ defined in Section 3.1.2, and $D_{i'}$ is the new target depth proposed by the LLM interpreter. Next, we shift and resize the synthesized depth map of this object to the target bounding box. A diffusion model is then called with ControlNet (Zhang et al., 2023) to generate an object with the specified depth. After generating a new object, the segmentation and backward diffusion are performed to obtain the latent representation of the object at the new depth. Finally, this latent representation is integrated into the image latent space to complete the modification.

## 4 EXPERIMENTS

### 4.1 DATASETS

**FoR-LMD**. We extend the LMD benchmark (Lian et al., 2024), which is a synthetic dataset and was designed to assess several reasoning skills that include spatial understanding. We augment the input spatial expressions in LMD by adding explicit perspective cues to incorporate FoR information. The LMD prompt template is: $(obj_1) (R_1)$ *and* $(obj_2) (R_2)$, where $obj_1$ and $obj_2$ are objects, and $R_1, R_2$ are spatial relations. We modify it to: $(obj_1) (R_1) (ref_1)$ *and* $(obj_2) (R_2) (ref_2)$, where $ref_1$ and $ref_2$ specify the reference perspective—camera view (relative), or object-centric view (intrinsic). To emphasize relations sensitive to perspective, we restrict $R_1, R_2$ to left, right, front, back. This results in 500 samples of spatial expression with explicit perspective.

**FoREST** (Premsri & Kordjamshidi, 2025) is a synthetic benchmark designed to evaluate the FoR understanding in multimodal models with FoR annotation. We sample 500 spatial expressions from the C-split of FoREST to match the size of FoR-LMD. Each prompt explicitly specifies the spatial perspective and the facing direction of the reference object, which is not provided in FoR-LMD.

### 4.2 EVALUATION METHOD

We adapted the proposed evaluation scheme in Wang et al. (2025b), which is shown to align with human judgment. However, we modified some evaluation aspects, such as facing direction. In detail, to evaluate the generated image, we call the Visual Perception module to extract the bounding boxes, depth, and orientations of key objects from an LLM parser as explained in Section 3.1. After obtaining the visual information for all key objects, we verify that the number of objects matches the given explanation in the text. We should note that in evaluated benchmarks, exactly one instance of each object must be present in the image. If this counting condition does not match, the image is considered incorrect. Next, we evaluate whether the detected orientation label matches the orientation specified in the annotated data. Any misalignment results in the image being marked as incorrect. Next, for the evaluation of the spatial relations, we consider the FoR annotation provided in the context. If the FoR is not camera-centric (relative), we convert the spatial relation into the camera perspective using the detected orientation of the reference object (relatum) by applying the same procedure explained in FoR Interpreter. Finally, we use the pre-defined geometric specifications of the spatial relations (Huang et al., 2023; Cho et al., 2023; Wang et al., 2025b), assuming the camera perspective, to assess the correctness of the spatial configuration. To validate our automatic

evaluation, we conducted a small human study on 200 randomly sampled images from the initial generation of FLUX.1 and 1 round of applying FoR-SALE. We measured agreement using Cohen's kappa. The resulting score of 0.791 indicates strong alignment between human judgments and our automatic evaluation. Details regarding this human study are in Appendix H.

### 4.3 BASELINE MODELS

For baseline comparison, we select six T2I models: Stable Diffusion (SD) 1.5(Rombach et al., 2022), SD 2.1(Rombach et al., 2022), SD 3.5-Large(Stability AI, 2024), GLIGEN(Li et al., 2023), FLUX.1(Black Forest Labs, 2025), and GPT-4o-image(OpenAI, 2025b). The number of Inference Steps is set to 30 for SD3.5-Large, recommended by the original paper (Stability AI, 2024), while the rest is set to 50. Other parameters are set to the default for all models. Given our focus on recent models, results for older baselines—including SD 1.5, SD 2.1, and GLIGEN—are presented in the Appendix. We also include a prompt-engineering approach for extracting a cognitive map Yang et al. (2025) as an additional baseline with GPT-4o, allowing comparison against more recent techniques. For comparison with editing frameworks that leverage LLMs to guide image modifications, we include SLD and GraPE Goswami et al. (2024). Both are self-correcting editing pipelines that achieve strong performance in spatial understanding by employing GPT-4o as the LLM interpreter. SLD serves as the original framework from which FoR-SALE is extended. GraPE, in contrast, is a general framework that leverages an MLLM to determine the appropriate editing actions and then invokes SOTA image editing models to modify the image according to the generated sequence of actions. All experiments were conducted on two A6000 GPUs, totaling around 400 GPU hours. Further implementation details of baseline models are provided in Appendix B.

### 4.4 FOR-SALE IMPLEMENTATION DETAIL

We select Qwen3-32B (Qwen Team, 2025) with reasoning enabled as the backbone for all LLM components used in the FoR-SALE pipeline. For the Visual Perception module, we employ OWLv2 (Minderer et al., 2024) for open-vocabulary object detection, DPT (Ranftl et al., 2021) for depth estimation, and OrientAnything (Wang et al., 2024b) for orientation detection. We utilize SD 1.5 as the base diffusion model for creating objects and the final step of denoising the composed latent space. The same visual processing tools are used in evaluation. To verify that our improvements are not tied to the performance of specific visual processing tools, we additionally report experimental results using alternative ones. The results demonstrate consistent improvements by the FoR-SALE framework independent of the choice of the vision tools. This outcome is expected, as the visual processing tools are limited to basic tasks (e.g., object detection), while the reasoning is carried out by the LLM. We report the results with the tools mentioned above. However, for the sake of sanity verification, we report the results with alternative tools in the Appendix D. Further implementation details of FoR-SALE are provided in Appendix A.

### 4.5 RESULTS

**RQ1. Can the SOTA T2I models follow the FoR expressed in the text?** As can be seen in Table 1, the best-performing model, GPT-4o, achieves only 52.20% accuracy, highlighting the difficulty of T2I generation—even with only two objects in a spatial relation. While GPT-4o performs well on relative FoR in FoR-LMD (94.76%), its accuracy drops sharply to 24.35% on intrinsic FoR, revealing a substantial performance gap. This trend is consistent with findings from FoREST(Premsri & Kordjamshidi, 2025) and GenSpace(Wang et al., 2025b), which emphasize the challenges of FoR reasoning beyond camera perspective. Interestingly, GPT-4o's advantage in relative FoR disappears in intrinsic settings, suggesting its improvements are largely limited to camera-based understanding. In the FoREST benchmark, which has explicit facing direction in the input, GPT-4o still maintains a relative lead—likely due to its better handling of facing direction. We also observe that GPT-4o may benefit from orientation cues in improving intrinsic FoR alignment. In contrast, other models fail to leverage such information and continue to struggle under both relative and intrinsic FoRs. In addition to the above baselines, we applied a spatial-prompting strategy that extracts a cognitive map based on Yang et al. 2025 with GPT-4o in an attempt to improve spatial reasoning. We found that the cognitive map improves GPT-4o's object-counting accuracy but still offers limited benefit for spatial relations with FoR. The detailed results are reported in Appendix C.3.

Table 1: Accuracy of generated images across baseline models and editing methods, including FoR-SALE. Relative denotes camera-based spatial expression; Intrinsic uses another object's perspective.

| Method | FoR-LMD | | | FoREST | | | Overall Avg. |
|---|---|---|---|---|---|---|---|
| | Relative | Intrinsic | Average | Relative | Intrinsic | Average | |
| SD 3.5 - Large | 63.75 | 24.72 | 42.60 | 18.11 | 11.11 | 15.00 | 28.80 |
| + 1-round GraPE | 55.46 | 16.97 | 34.60 | 14.91 | 7.56 | 11.60 | 23.10 |
| + 1-round SLD | 61.57 | 19.56 | 38.80 | 22.55 | 11.55 | 17.60 | 28.20 |
| **+ 1-round FoR-SALE (Ours)** | 61.14 | 26.56 | 42.40 | 24.00 | 16.00 | 20.40 | 31.40 |
| **+ 2-round FoR-SALE (Ours)** | 67.25 | 26.94 | 45.40 | 28.00 | 22.22 | 25.40 | 35.40 |
| **+ 3-round FoR-SALE (Ours)** | **70.31** | **29.52** | **48.20** | **28.00** | **22.22** | **25.40** | **36.80** |
| FLUX.1 | 58.95 | 25.83 | 41.00 | 18.18 | 15.56 | 17.00 | 29.00 |
| + 1-round GraPE | 54.15 | 18.08 | 34.60 | 17.45 | 11.56 | 14.80 | 24.70 |
| + 1-round SLD | 63.32 | 25.09 | 42.60 | 24.72 | 12.00 | 19.00 | 30.80 |
| **+ 1-round FoR-SALE (Ours)** | 65.07 | 27.67 | 44.80 | 25.09 | 22.22 | 23.80 | 34.30 |
| **+ 2-round FoR-SALE (Ours)** | 67.68 | **28.04** | **46.20** | 30.18 | 29.78 | 30.00 | 38.10 |
| **+ 3-round FoR-SALE (Ours)** | **69.43** | 25.84 | 45.80 | **32.72** | **31.11** | **32.00** | **38.90** |
| GPT-4o | 94.76 | 24.35 | 56.60 | **57.81** | 35.56 | 47.80 | 52.20 |
| + 1-round GraPE | 93.89 | 19.56 | 53.60 | 55.64 | 30.22 | 44.20 | 48.90 |
| + 1-round SLD | 89.08 | 21.40 | 52.40 | 43.27 | 23.56 | 34.40 | 43.40 |
| + Cognitive Map | **98.23** | 30.36 | 62.00 | 56.00 | 37.33 | 47.60 | 54.80 |
| **+ 1-round FoR-SALE (Ours)** | 93.01 | 35.42 | 61.80 | 54.18 | 37.33 | 46.60 | 54.20 |
| **+ 2-round FoR-SALE (Ours)** | 93.01 | 34.32 | 61.20 | 48.73 | 39.11 | 44.40 | 52.80 |
| **+ 3-round FoR-SALE (Ours)** | 91.26 | **38.37** | **62.60** | 53.81 | **42.22** | **48.60** | **55.60** |

**RQ2. How effective is FoR-SALE framework in editing images to follow the FoR expressed in text?** To answer this question, we compare FoR-SALE with two existing auto-editing frameworks: SLD and GraPE. FoR-SALE generally outperforms both, except in the relative FoR setting of the FoR-LMD benchmark, where SLD slightly excels. We attribute this to the simplicity of camera perspective contexts in that setting, which do not require FoR reasoning. However, FoR-SALE is still competitive with only a minor 0.40% accuracy drop. In contrast, for more challenging intrinsic FoR settings, FoR-SALE achieves substantial improvement, up to 5% after one round and 15% after three rounds. Other frameworks consistently struggle in such cases. We also observe consistent overall performance improvements with additional rounds of FoR-SALE. Figure 4 presents a detailed error analysis comparing images from FLUX.1 with those edited by SLD and FoR-SALE. FoR-SALE shows clear improvements in left and right relations, which can often be corrected through 2D spatial adjustments. These gains are expected when the layout interpreter accurately infers the FoR, highlighting the positive impact of the FoR Interpreter. It also reduces many orientation errors, though correcting 3D aspects such as depth and facing direction remains challenging, with a high error rate in those categories. Performance on front and back relations shows limited improvement and sometimes worsens compared to SLD, underscoring the difficulty of 3D editing. We suspect that SLD's apparent improvement in front/back errors does not lead to an overall performance increase, as it introduces new errors due to a lack of depth information. To evaluate this hypothesis, we provide a detailed analysis in Appendix E, comparing errors in front and back relations. The analysis reveals that SLD's front/back errors are reduced due to the generation of extra objects, which are later counted as multiple-object errors. We also observe that multiple-object and missing-object errors remain high for both models, highlighting a limitation in current editing frameworks. Finally, by sampling failure cases and manually categorizing each error, we find that the majority of mistakes arise from incorrect orientation generation, failures in the final diffusion stage to synthesize the target object, and shortcomings of the Visual Perception Modules in detection. Further details of this are provided in Appendix F.

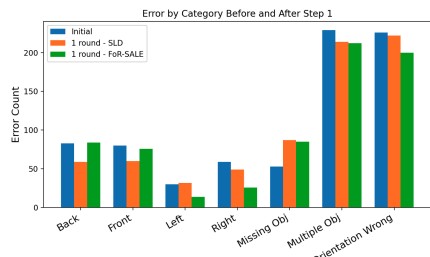

Figure 4: Error analysis of images generated by FLUX.1 (blue) and after one round of editing using SLD (orange) or FoR-SALE (green).

Table 2: Accuracy of suggested layout and edited images from the corresponding layout under different Layout Interpreters using initial images generated from GPT4o.

| | LLM-Layout Accuracy | | | Image Accuracy | | |
|---|---|---|---|---|---|---|
| Layout Interpreters | Relative | Intrinsic | Average | Relative | Intrinsic | Average |
| o3 | **99.40** | 79.03 | **89.30** | 69.24 | 30.64 | 50.10 |
| o4-mini | 99.20 | 64.52 | 82.00 | **74.40** | 29.44 | 52.10 |
| Qwen3 | 98.21 | 45.97 | 72.30 | 73.61 | 21.77 | 47.90 |
| FoR-Interpreter(No-Rules) + Qwen3 | 95.23 | 54.03 | 74.80 | 69.84 | 24.80 | 47.50 |
| FoR-Interpreter(Partial-Rules) + Qwen3 | 93.25 | 81.65 | 87.50 | 70.63 | **39.52** | **55.20** |
| FoR-Interpreter(Full-Rules) + Qwen3 | 93.85 | **84.48** | 89.20 | 71.82 | 36.29 | 54.20 |

Table 3: Accuracy of image generated from FoR-SALE for two rounds with exclude either facing or depth Modification and SLD using initial images generated from FLUX.1.

| | Accuracy | | |
|---|---|---|---|
| Method | Relative | Intrinsic | Average |
| FLUX.1 | 38.57 | 20.70 | 29.00 |
| + SLD | 42.26 | 19.15 | 30.80 |
| + FoR-SALE | 43.25 | 25.20 | 34.30 |
| - Facing Direction Modification | 40.67 | 22.17 | 31.50 |
| - Depth Modification | 42.65 | 25.20 | 34.00 |
| + FoR-SALE (2-round) | 47.22 | 28.83 | 38.10 |
| - Facing Direction Modification | 45.83 | 26.21 | 36.10 |

## 5 ABLATION STUDY

**RQ3. How accurate do the LLMs perform Layout-Editing?** To address this question, we conduct an ablation study on the LLMs used for the Layout Interpreter, evaluating two SOTA reasoning models: o3 and o4-mini (OpenAI, 2025a). We also examine three settings for the FoR Interpreter. (1) No-Rule, where no rules are provided. (2) Partial-Rules, which include only facing direction-related rules explicitly present in the input or detection results. (3) Full-Rules, which include all rules. We report accuracy using the evaluation protocol described in Section 4.2, measuring the quality of the LLM-generated layout and the accuracy of the final image produced after editing. Table 2 presents the results of this experiment. The accuracy of the LLM-generated layouts is significantly higher than that of the corresponding generated images, highlighting the challenge of correctly executing layout-guided edits. Despite this, a clear performance gap remains between relative (camera-centric) and intrinsic (non-camera) FoR—particularly for Qwen3 without the FoR Interpreter. We observe that incorporating the FoR Interpreter leads to noticeable performance improvements for Qwen3, especially in handling intrinsic FoR. Moreover, adding perspective-conversion rules further enhances Qwen3's intrinsic FoR reasoning. The improvement is substantial, even when only a partial set of rules is provided. This highlights that such rules—despite being incomplete—substantially assist the model in performing perspective conversion, as evidenced by significant gains over the Qwen3 baseline. Notably, with these enhancements, Qwen3 outperforms o3 on intrinsic FoR, which presents the more challenging reasoning. Although the FoR Interpreter slightly reduces Qwen3's layout accuracy in the relative case (by 5%), it yields a substantial +38.5% improvement on intrinsic FoR, affirming the overall effectiveness of this module. We also find that although o3 produces more accurate layouts than both o4-mini and our layout interpreter, it results in a lower final image accuracy. We hypothesize that this is due to o3's generated layouts requiring a higher number of editing actions, making it more difficult for the editing framework. To evaluate this hypothesis, we analyze the distribution of editing actions required to align the image with the newly generated layout. Our analysis shows that o3's layouts require, on average, more repositioning operations and a higher number of total actions than those generated by the other LLMs; the details are reported in Appendix C.

**RQ4. How do the new editing actions help FoR-SALE?** To answer this question, we conduct an ablation study by disabling facing direction or depth modification in FoR-SALE, using initial images from FLUX.1. As shown in Table 3, removing facing direction modification reduces accuracy by

2.8%, while removing depth modification leads to a 0.30% drop. A minor 0.3% accuracy difference highlights the gain from the first round of editing only. This difference could potentially be higher when we use more rounds of editing with FoR-SALE. We provide the results of applying FoR-SALE without the Depth Modification operation in the second round in the same table, where the overall accuracy drops by 2%. This shows that the Depth Modification still has an impact during further refinement, even if it appears to have less influence in the first round. All of these results highlight the importance of both editing actions—depth editing and facing direction—in improving the spatial alignment of our framework. The limited impact of depth editing in the first round suggests it remains a challenge, and future work may focus on enhancing its effectiveness.

**RQ5. How does incorporating FoR-SALE impact the execution time of the image generation?**

To address this question, we provide the execution time for each component in the Table 4, averaged over 1 round of applying FoR-SALE to the initial image generated by FLUX.1. We also provide the execution time for generating an image with FLUX.1 as the baseline. All times are recorded on 1 GPU A6000 with 48GB. We observe up to 2.5x more time compared to the baseline image generation (FLUX.1) when using FoR-SALE. Three components contribute most to the overhead, all of which are LLM-based. The LLM Parser used to extract objects from the sentence will be called only in the first round and will not add any overhead to the next rounds of editing. The Layout Interpreter is the most time-consuming component, as the LLM must process and propose the layout. We believe this cost is comparable to the original SLD framework, which also includes these two components (i.e., Parser and interpreter). The new component introduced by our method, that is the FoR Interpreter, increases the execution time by only around 0.5x relative to the baseline we built upon. This trade-off is a reasonable choice for enabling the framework to understand spatial relations from non-camera perspectives—a challenging aspect that current methods largely overlook—and achieve a better accuracy (5% when applying 1 round of FoR-SALE) compared to SLD under our evaluation benchmarks.

Table 4: Execution time of each component in FoR-SALE compared with the average image-generation time of FLUX.1 under the same settings.

| Component | Time (s) |
|---|---|
| FLUX.1 | 49.52 |
| LLM Parser | 21.64 |
| Object Detection | 0.94 |
| Depth Detection | 1.23 |
| Orientation Detection | 1.60 |
| FoR Interpreter | 31.84 |
| Layout Interpreter | 45.52 |
| Latent Operations | 17.91 |
| Generate new image | 7.99 |
| Avg. | 128.67 |

## 6    Conclusion

Given the limitations of current text-to-image (T2I) models in handling spatial relations across diverse frames of reference (FoR), we propose FoR-SALE—Frame of Reference-guided Spatial Adjustment in LLM-based Diffusion Editing—to address this challenge. Our framework extends the Self-correcting LLM-controlled Diffusion approach by introducing three key components: a comprehensive Visual Perception Module, a dedicated FoR Interpreter, and two new latent editing actions. FoR-SALE can be seamlessly integrated into various T2I models and effectively improves the spatial alignment of images initially generated by those models—achieving up to 5.30% improvement in a single correction round and 9.90% in 3 rounds. Using GPT-4o as the base generator, our method achieves SOTA performance on spatial expressions involving FoRs, particularly for intrinsic FoRs, which are especially challenging. These results demonstrate the robustness of reasoning over FoR of our proposed framework.

## Ethics Statement

While we identify shortcomings of existing Text-to-Image models, our intention is to highlight areas for improvement rather than to disparage prior work. Our analysis is constrained to a synthetic environment that provides controlled conditions but may not fully capture real-world contexts. In addition, our study is limited to English and does not account for linguistic or cultural variations in spatial expression. Extending this work to multiple languages may reveal important differences in frame-of-reference comprehension. We emphasize that these modules are used solely for com-

parative purposes and do not resolve the broader challenges of visual perception. Large language models were also used to assist with grammar checking, sentence refinement, and the search for some related works. Finally, our experiments require substantial GPU resources, which restricted the range of large language models we were able to test. These computational demands also pose accessibility challenges for researchers with limited resources.

## REPRODUCIBILITY STATEMENT

To ensure the reproducibility of our experiments, we will release the code in a public repository upon acceptance, along with all datasets created for this work. We provide detailed implementation settings for FoR-SALE and all baseline models, including hyperparameters and other configurations, in Section 4 and Appendix A. All baseline implementations used in our experiments are publicly available, and we rely on either official releases or widely adopted open-source repositories to maintain consistency and comparability.

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

## A    FoR-SALE Implementation Details

Random seed are set into an arbitrary number, 78 in all of our experiments, for reproducible results.

### A.1    LLM Parser

For the implementation of the LLM Parser, we employ Qwen3-32B with reasoning generation (thinking tokens) disabled to enable faster inference, given the simplicity of the task. The temperature is set to 0 for reproducible results, and the maximum token limit is 8196. Listing 3 in Section I provides the complete prompt and examples used for this LLM Parser.

### A.2    FoR Interpreter

We select Qwen3-32B with reasoning generation (thinking tokens) enabled for the FoR Interpreter, as this component requires reasoning over the provided rules. To ensure reproducibility, the temperature is set to 0, and the maximum token limit is 8196. Listing 6 in Section I presents the complete prompt and examples used for the FoR Interpreter.

### A.3    Layout Interpreter

Similar to the FoR Interpreter, we use Qwen3-32B with reasoning generation (thinking tokens) enabled for this LLM component. For the ablation study, we also evaluate two additional LLMs via the OpenAI API: o3 (model name: o3-2025-04-16) and GPT-o4-mini, both from OpenAI. To ensure reproducibility, the temperature is set to 0, and the maximum token limit is 8196. This configuration is applied consistently across all LLMs used in the Layout Interpreter. The prompt for this Layout Interpreter is in Listing 6 in Section I.

### A.4    Visual Perception Module

For the implementation of the Visual Perception Module, we employ three components including object detection, depth estimation, and orientation detection as mentioned in the main paper. For open-vocabulary object detection, we use OWLViT2, with the model ID *google/owlv2-base-patch16-ensemble*. For depth estimation, we select DPT, using the model ID *Intel/dpt-large*. Finally, for orientation detection, we employ OrientAnything, with ViT-Large as the base model. The model weights are loaded from the checkpoint *croplargeEX2/dino_weight.pt*, as provided in the official GitHub repository.

### A.5    Evaluation Functions

There are a total of four evaluation functions used to evaluate the generated image. The visual details are represented in the following format: ((attribute) (object name) (#object

`ID)`, $[x, y, w, h]$, $D_i$, $f_i$) where $(x, y)$ indicates the coordinates of the upper-left corner of the bounding box from 0.0 to 1.0, $w$ is its width, $h$ is its height, $D_i$ is depth from 0.0 to 1.0 which 1.0 is indicate nearest to the camera, and $f_i$ is facing direction label. Each comparison involves two objects, denoted as $obj_1$ and $obj_2$. Before performing the comparison, we compute the center of each object's bounding box, denoted by $(c_x, c_y)$, where $c_x = x + w/2$ and $c_y = y + h/2$. The procedure for each comparison is described below.

- **Left**. We determine whether the center of $obj_1$ is to the left of $obj2$ by checking whether $c_x$ of $obj_1$ is less then $c_x$ of $obj2$. The condition is defined as,

$$c_x^{obj_1} < c_x^{obj2}$$

.

- **Right**. We determine whether the center of $obj_1$ is to the right of $obj2$ by checking whether $c_x$ of $obj_1$ is greater then $c_x$ of $obj2$. The condition is defined as,

$$c_x^{obj_1} > c_x^{obj2}$$

- **Front**. We determine whether $obj_1$ is front of $obj_2$ by comparing $D_1$ (depth of $obj1$) with $D_2$ (depth of $obj2$) . The condition is defined as,

$$D_1 > D_2$$

- **Back**. Similar to front relation, we compare $D_1$ with $D_2$ using following condition,

$$D_1 < D_2$$

## B    BASELINE MODELS PARAMETERS

### B.1    STABLE DIFFUSION (SD)

For baselines using SD1.5 and SD2.1, we set the number of inference steps to 50, while keeping all other parameters at their default values. The model ID for SD1.5 is *sd-legacy/stable-diffusion-v1-5*, and for SD2.1, it is *stabilityai/stable-diffusion-2-1*. The baseline using SD3.5-Large employs the model ID *stabilityai/stable-diffusion-3.5-large*, with the number of inference steps set to 30; all other parameters remain unchanged.

### B.2    GLIGEN

We use Qwen3 to generate the initial layout for the GLIGEN baseline. The prompt used for layout generation is shown in Listing 2. For the GLIGEN model, we use the model ID *masterful/gligen-1-4-generation-text-box*. We also provide facing direction information when generating images with GLIGEN by augmenting the object names with the corresponding facing directions extracted from the layout generated by Qwen3. The number of inference steps is set to 50, while all other parameters remain unchanged.

### B.3    FLUX.1

For generating images with FLUX.1 baseline, we employ the pipeline with model id *black-forest-labs/FLUX.1-dev*. The guidance scale is set to 3.5, following the recommended value. The image resolution is 1024×1024, and the number of inference steps is set to 50. Other parameters are set as default.

### B.4    GPT4O-IMAGE

We utilize the OpenAI API to generate images for the GPT-4o baseline, employing the model ID gpt-image-1. The background setting is set to auto, and the image resolution is configured to 1024×1024. All other parameters are left at their default values. The cost for generating one image is around $\$0.01 - \$0.02$.

Table 5: Accuracy of generated images across pioneer diffusion models and editing methods.

| Method | FoR-LMD | | | FoREST | | | Overall Avg. |
|---|---|---|---|---|---|---|---|
| | Relative | Intrinsic | Average | Relative | Intrinsic | Average | |
| SD 1.5 | 12.66 | 11.80 | 12.20 | 7.63 | 4.00 | 6.00 | 9.10 |
| SD 2.1 | 13.97 | 10.33 | 12.00 | 5.09 | 7.11 | 6.00 | 9.00 |
| Qwen3 + GLIGEN | 58.52 | 21.40 | 38.40 | 2.54 | 1.33 | 2.00 | 20.20 |

Table 6: The percentage of editing action required for editing both FoR-LMD and FoREST using the initial image from GPT4o based on different Layout Interpreters. FoR-I stands for FoR-Interpreter. Attribute refers to Attribute Modification, Depth refers to Depth Modification, and Facing refers to Facing Direction Modification.

| Layout Interpreter | Add | Remove | Attribute | Reposition (R) | Facing | Depth (D) | D + R | # Actions |
|---|---|---|---|---|---|---|---|---|
| o3 | 3.60 | 10.63 | 0.00 | 49.82 | 15.95 | 10.45 | 9.55 | 1110 |
| o4-mini | 4.98 | 11.92 | 0.00 | 39.85 | 20.64 | 18.98 | 3.75 | 906 |
| Qwen3 | 3.66 | 10.47 | 0.00 | 36.65 | 16.86 | 25.65 | 6.70 | 955 |
| FoR-I(No-Rules)+Qwen3 | 3.81 | 9.18 | 0.00 | 42.47 | 13.40 | 26.91 | 4.23 | 970 |
| FoR-I(Partial-Rules)+Qwen3 | 3.33 | 8.22 | 0.00 | 41.78 | 10.76 | 31.12 | 4.79 | 1022 |
| FoR-I(Full-Rules)+Qwen3 | 3.40 | 6.90 | 0.00 | 43.25 | 11.95 | 29.74 | 4.86 | 1029 |

# C  ADDITIONAL RESULTS ON TEXT-TO-IMAGE (T2I) BASELINES

## C.1  ADDITIONAL RESULTS OF PIONEER T2I

We provide additional results for early T2I models, including SD1.5, SD2.1, and GLIGEN, using layouts generated by Qwen3, as shown in Table 5. All models perform significantly worse than the SOTA baselines discussed in the main results—particularly SD1.5 and SD2.1, which achieve less than 10% accuracy. While GLIGEN shows more acceptable performance on the FoR-LMD benchmark, it performs poorly when orientation requirements are introduced, as in the context of the FoR-EST benchmark. GLIGEN's accuracy drops to just 2%, indicating a lack of understanding of object-level attributes—especially facing direction—even when this information is explicitly provided during generation.

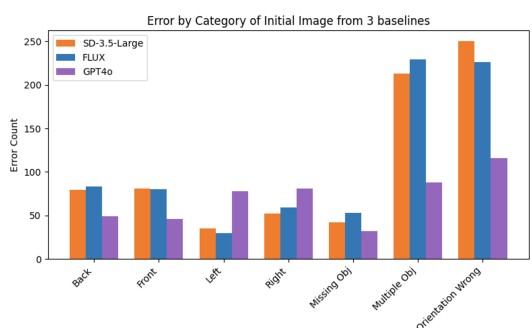

Figure 5: Error analysis of image generated by SD-3.5-Large, FLUX.1, and GPT-4o,

## C.2  IMAGE GENERATION ERROR OF DIFFERENT BASELINES

Figure 5 illustrates the error distribution for images generated by SD3.5-Large, FLUX.1, and GPT-4o. We observe notable differences among these models. Note that, while SD3.5-Large and FLUX.1 are diffusion-based T2I models, GPT-4o is a unified generative model trained on multimodal input-output tasks. GPT-4o exhibits significantly fewer missing or additional key objects, indicating stronger object grounding and a more accurate object count. It also shows lower error rates in front/back relations and orientation, suggesting improved performance in handling 3D spatial configurations, including depth and facing direction. However, GPT-4o performs worse on left/right relations compared to the diffusion-based models. We anticipate that this may be attributed to challenges in perspective conversion, as evidenced by GPT-4o's high performance on relative FoRs in the FoR-LMD benchmark (94.76%), which requires only camera-centric understanding, contrasted with its significantly lower accuracy on intrinsic FoRs, as reported in the main results. These findings suggest a trade-off in GPT-4o's spatial performance—namely, strong handling of camera-centric spatial expressions, but limited generalization to non-camera perspectives in text-to-image tasks.

Table 7: Accuracy of generated images with cognitive map and FoR-SALE. Relative denotes camera-based spatial expression; Intrinsic uses another object's perspective.

| Method | FoR-LMD | | | FoREST | | | |
| | Relative | Intrinsic | Average | Relative | Intrinsic | Average | Overall Avg. |
|---|---|---|---|---|---|---|---|
| GPT-4o | **94.76** | 24.35 | 56.60 | **57.81** | 35.56 | 47.80 | 52.20 |
| + Cognitive Map | 98.23 | 30.36 | 62.00 | 56.00 | 37.33 | 47.60 | 54.80 |
| **+ 1-round FoR-SALE (Ours)** | 93.01 | 35.42 | 61.80 | 54.18 | 37.33 | 46.60 | 54.20 |

Table 8: Number of errors in each category obtained using the cognitive map and FoR-SALE on the initial images generated by FLUX.1.

| Error Type | Missing Object | Over-Generated Objects | Spatial Relations |
|---|---|---|---|
| Cognitive Map | 7 | 38 | 173 |
| FoR-SALE | 10 | 75 | 177 |

## C.3 COGNITIVE MAP

We provide additional results from applying the current SpatialLM, that is, cognitive map prompting Yang et al. (2025). We use prompt engineering to extract a cognitive map before generating the image with GPT-4o, compared to 1 round of applying FoR-SALE. We provide the results in Table 7. We observe that the model achieves a significant gain on FoR-LMD, especially on the relative FoR. This is expected, as the cognitive map should improve 2D spatial relations. We further compared the errors in FoR-LMD based on the output of our automatic evaluator to investigate the source of this improvement in Table 8. We found that the cognitive map helps the model predict a more accurate number of objects, which contributes to the improvement of FoR-LMD in both the relative and intrinsic FoR. However, this model with the cognitive map still struggles with FoR reasoning, leading to more errors in spatial relations. This is where our proposed framework provides the most value.

## D VISUAL PERCEPTRON MODULES

In this section, we present additional experiments with alternative stacks of visual perception modules to ensure that our improvements are not tied to the specific evaluation modules used in FoR-SALE or the evaluation protocol. Specifically, we replace object detection with Grounding-DINOLiu et al. (2023b) and depth estimation with MiDaS 3.0Birkl et al. (2023). We then evaluate the same images generated with FLUX and a single round of FoR-SALE, using the initial images from FLUX as input. The results are reported in Table 9. We observe that both stacks of evaluation modules yield a similar improvement of approximately 4% over the initial images, although there is a minor variation of about 2% between the two results. These findings indicate that FoR-SALE consistently improves performance, even when different stacks of visual perception modules are employed.

## E ANALYSIS OF FOR-SALE FRAMEWORK

### E.1 ADDITIONAL QUANTITATIVE ERROR ANALYSIS OF FOR-SALE

In this section, we further analyze the errors observed after applying one round of FoR-SALE to the initial images generated by FLUX.1 We compare SLD and FoR-SALE in editing images containing front/back spatial relation errors in Figure 6. We observe that while SLD attempts to correct the front/back relation, it often introduces multiple instances of the target objects instead of editing the original ones. This behavior results in a lower front/back error after one round of editing, but it comes at the cost of generating additional object-related errors. We attribute this limitation to SLD's lack of depth awareness, which leads to incorrect editing operations. In contrast, FoR-SALE, which incorporates depth information, achieves slightly better correction on front/back errors without in-

Table 9: Accuracy of generated images across different stacks of visual perception modules in the evaluation protocol.

| Method | Main Visual Perception Modules | | | Alternative Visual Perception Modules | | |
|---|---|---|---|---|---|---|
| | Relative | Intrinsic | Average | Relative | Intrinsic | Average |
| FLUX.1 | 36.70 | 21.17 | 29.00 | 38.69 | 23.39 | 31.10 |
| + 1-round FoR-SALE | 43.25 | 25.20 | 34.30 | 41.07 | 30.44 | 35.80 |

troducing new object duplication or misalignment. Importantly, FoR-SALE avoids introducing new error types, making it more robust for subsequent editing rounds.

### E.2 DETAIL ANALYSIS OF THE EFFECT OF DIFFERENT LAYOUT INTERPRETERS AND EDITING ACTIONS

We report the distribution of editing actions required for images generated by GPT-4o when using different Layout Interpreters in Table 6. We observe that o3 requires significantly more editing actions compared to other models, with repositioning accounting for 59.37% of all actions (repositioning and depth modification with repositioning). This suggests that o3 often generates layouts where the object is repositioned, likely indicating that it is proposing an entirely new scene layout rather than minimally adjusting the original. This behavior may explain the performance drop observed when using o3-generated layouts, as reported in the main results. It also highlights a limitation of the FoR-SALE framework, the difficulty in handling cases that require multiple or complex repositioning actions. These findings suggest that future work may explore improved strategies for accurately moving objects—or even fully regenerating images—when layout revisions are extensive.

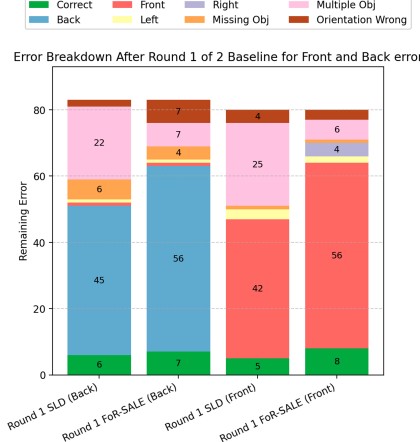

Figure 6: Error breakdown after one round of editing initial images from FLUX.1 using SLD and FoR-SALE on front and back relation errors.

## F FAILURE CASE OBSERVATION

We present examples of FoR-SALE editing failures in Figure 7. The most common errors include multiple instances of key objects, incorrect orientation, and missing objects, as also reflected in the main paper's quantitative results. We anticipate these failures primarily to challenges in object removal and re-generation, which can lead to either the unintended deletion of key objects or the generation of extraneous ones—ultimately making the intended objects undetectable in the final image. Additionally, we believe that modifying orientation and depth remains difficult for current diffusion models, which limits the effectiveness of FoR-SALE in correcting these types of spatial errors.

### F.1 PERCENTAGE OF FOR-SALE FAILURE CASES

**Observation Setting.** To identify the sources of FoR-SALE generation errors, we sample 60 images (10% of failure cases in round 1) from the incorrect cases produced by applying one round of FoR-SALE to initial images generated by FLUX.1. We categorize the errors into six types, including

**1. Removing object failure (E1).** FoR-SALE fails to remove an object from the image completely.

**2. Incorrect orientation generation (E2).** The diffusion model in FoR-SALE generates objects with incorrect orientations.

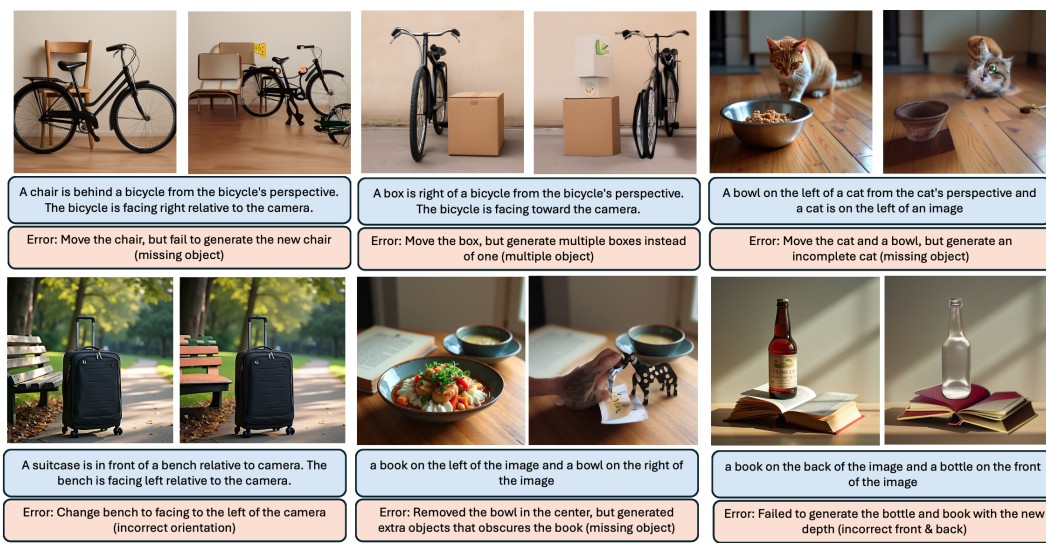

Figure 7: Examples of editing errors using FoR-SALE. The blue box indicates the input spatial expression, while the orange box explains the editing action and the underlying reason for the error.

**3. Incorrect depth generation (E3).** The diffusion model with ControlNet in FoR-SALE generates images with incorrect depth.

**4. Incorrect object generation (E4).** The final Stable Diffusion model in FoR-SALE fails to synthesize the correct object image from the edited latent space.

**5. Object detection failure (E5).** The visual perception modules fail to detect correct object properties or spatial coordinates.

**6. LLM failure (E6).** The LLM-controlled editing suggests an incorrect image layout, leading to erroneous edits.

**Results.** We report the manually checked cases in Figure 8. The bar chart illustrates the percentage of each error that occurs in the incorrect sample. We observe three major errors that contributed to the failure of FoR-SALE. The first error(23.33% of all errors) is generation failure, which occurs when multiple compositions of the latent space overlap at the same pixel location, leading to poor-quality object images (example in the upper-left picture in

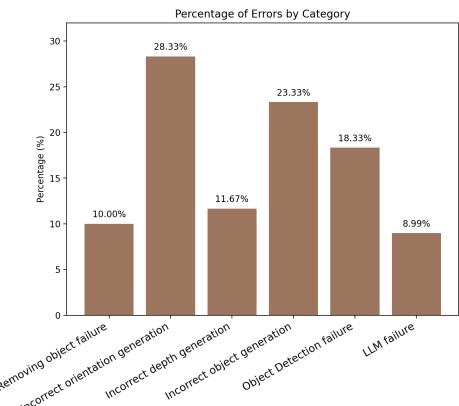

Figure 8: Percentage of each error category from 60 samples of failure cases of applying 1 round of FoR-SALE to initial images from FLUX.1.

Figure 7). This issue is closely related to the second source of error, object detection failure (18.33% of all errors), which arises when two objects are positioned too closely. In such cases, the detector may fail to capture all objects or may instead focus on incorrect regions of the image, resulting from flawed generation. The third source of error and the most significant one is incorrect orientation generation (28.33% of all errors). As noted in Wang et al. 2025b, even SOTA T2I models struggle to produce correct object orientations, highlighting a fundamental challenge that requires stronger 3D awareness in diffusion models. A similar challenge is observed for depth generation, where limited progress has been made in depth-editing actions. To provide further insight into DiffEdit's orientation editing, we examined the images generated by DiffEdit in 1 round of FoR-SALE across three baselines that solely edited the orientation of a single object. We report the error percentage of DiffEdit orientation edits in Table 10. The soft-error metric allows more orientation labels to be considered correct (e.g., treating "left" as correct for

backward-left and forward-left, and vice versa), whereas the hard-error metric requires the generated orientation to match the expected label exactly.

# G  Perspective Conversion Rules

In this section, we present all perspective conversion rules used in the FoR Interpreter and the corresponding evaluation method. The rules are categorized by the facing direction of the reference object. Each facing direction is associated with exactly four conversion rules, corresponding to the four spatial relations considered in this work, i.e., left, right, front, and back.

Table 10: The error percentage of DiffEdit for editing orientation based on applying 1 round of FoR-SALE across three baselines.

| Baseline | Soft-Error | Hard-Error |
|---|---|---|
| SD 3.5 | 56.25 | 73.20 |
| FLUX.1 | 67.10 | 78.71 |
| GPT-4o | 49.47 | 72.35 |

1. Facing toward the camera.

   (a) Left. If the object is facing toward the camera (front), then the left side of the object is on the right from the camera perspective.
   (b) Right. If the object is facing toward the camera (front), then the right side of the object is on the left from the camera perspective.
   (c) Front. If the object is facing toward the camera (front), then the front side of the object is in the front direction from the camera perspective.
   (d) Back. If the object is facing toward the camera (front), then the back side of the object is in the back direction from the camera perspective.

2. Facing forward-left.

   (a) Left. If the object is facing forward-left, then the left side of the object is on the right from the camera perspective.
   (b) Right. If the object is facing forward-left, then the right side of the object is on the left from the camera perspective.
   (c) Front. If the object is facing forward-left, then the front side of the object is in the front direction from the camera perspective.
   (d) Back. If the object is facing forward-left, then the back side of the object is in the back direction from the camera perspective.

3. Facing left.

   (a) Left. If the object is facing left, then the left side of the object is in the front direction from the camera perspective.
   (b) Right. If the object is facing left, then the right side of the object is in the back direction from the camera perspective.
   (c) Front. If the object is facing left, then the front side of the object is on the left from the camera perspective.
   (d) Back. If the object is facing left, then the back side of the object is on the right from the camera perspective.

4. Facing backward-left.

   (a) Left. If the object is facing backward-left, then the left side of the object is on the left from the camera perspective.
   (b) Right. If the object is facing backward-left, then the right side of the object is on the right from the camera perspective.
   (c) Front. If the object is facing backward-left, then the front side of the object is in the back direction from the camera perspective.

(d) Back. If the object is facing backward-left, then the back side of the object is in the front direction from the camera perspective.

5. Facing away from the camera.

   (a) Left. If the object is facing away from the camera (back), then the left side of the object is on the left from the camera perspective.

   (b) Right. If the object is facing away from the camera (back), then the right side of the object is on the right from the camera perspective.

   (c) Front. If the object is facing away from the camera (back), then the front side of the object is in the back direction from the camera perspective.

   (d) Back. If the object is facing away from the camera (back), then the back side of the object is in the front direction from the camera perspective.

6. Facing backward-right.

   (a) Left. If the object is facing backward-right, then the left side of the object is on the left from the camera perspective.

   (b) Right. If the object is facing backward-right, then the right side of the object is on the right from the camera perspective.

   (c) Front. If the object is facing backward-right, then the front side of the object is in the back direction from the camera perspective.

   (d) Back. If the object is facing backward-right, then the back side of the object is in the front direction from the camera perspective.

7. Facing right.

   (a) Left. If the object is facing right, then the left side of the object is in the back direction from the camera perspective.

   (b) Right. If the object is facing right, then the right side of the object is in the front direction from the camera perspective.

   (c) Front. If the object is facing right, then the front side of the object is on the right from the camera perspective.

   (d) Back. If the object is facing right, then the back side of the object is on the left from the camera perspective.

8. Facing forward-right.

   (a) Left. If the object is facing forward-right, then the left side of the object is on the right from the camera perspective.

   (b) Right. If the object is facing forward-right, then the right side of the object is on the left from the camera perspective.

   (c) Front. If the object is facing forward-right, then the front side of the object is in the front direction from the camera perspective.

   (d) Back. If the object is facing forward-right, then the back side of the object is in the back direction from the camera perspective.

# H  HUMAN ALIGNMENT SCORE

Listing 1: Insturction for collecting human results on evaluating generated images.

```
"""
Instruction:
You will be presented with an image generated by a generative model along
    with its corresponding input prompt. Your task is to determine
    whether the generated image aligns with the prompt. If the image is
    aligned, assign a score of 1; otherwise, assign a score of 0.
"""
```

## H.1 EXPERIMENTAL SETTING

We provide a set of 200 images—sampled from the initial FLUX.1 generations and the first round of FoR-SALE—to a human annotator, compensated as a research assistant. This selection ensures diversity in the generated outputs and maintains independence from our specific generation framework. The annotator follows the instructions in Listing 1 to judge whether each image aligns with the corresponding input description. Each sample is labeled as either aligned (1) or not aligned (0), and we collect these binary scores for all images.

## I    PROMPT SPECIFICATIONS

We provide the prompt for LLM used throughout the entire experiments in this section.

Listing 2: Prompt for generate layout for GLIGEN.

```
Your task is to generate the bounding boxes of objects mentioned in the
    caption, along with direction that objects facing.
The image is size 512x512.
The bounding box should be in the format of (x, y, width, height) from 0
    to 1.
The direction that object is facing should be one of these options, [
    front, back, left, right]
Please considering the frame of reference of caption and direction of
    reference object.
The answer should be in the form of "Reasoning: Explanation\nLayout:
    Layout\" The example of layout is [(cat, [0.1, 0.3, 0.5, 0.4], right)
    , (cow, [0.6, 0.5, 0.3, 0.4], right)]"
```

Listing 3: Prompt for LLM Parser.

```
# Your Role: Excellent Parser

## Objective: Analyze scene descriptions to identify objects and their
    attributes.

## Process Steps
1. Read the user prompt (scene description).
2. Identify all objects mentioned with quantities.
3. Extract attributes of each object (color, size, material, etc.).
4. Ignore facing attribute (facing to left, facing to right, facing
    forward)
5. If the description mentions objects that shouldn't be in the image,
    take note at the negation part.
6. Explain your understanding (reasoning) and then format your result (
    answer / negation) as shown in the examples.
7. Importance of Extracting Attributes: Attributes provide specific
    details about the objects. This helps differentiate between similar
    objects and gives a clearer understanding of the scene.

## Examples

- Example 1
    User prompt: A brown horse is beneath a black dog. Another orange cat
        is beneath a brown horse.
    Reasoning: The description talks about three objects: a brown horse,
        a black dog, and an orange cat. We report the color attribute
        thoroughly. No specified negation terms. No background is
        mentioned and thus fill in the default one.
    Objects: [('horse', ['brown']), ('dog', ['black']), ('cat', ['orange
        '])]
    Background: A realistic image
    Negation:

- Example 2
```

```
     User prompt: There's a white car and a yellow airplane in a garage.
         They're in front of two dogs and behind a cat. The car is small.
         Another yellow car is outside the garage.
     Reasoning: The scene has two cars, one airplane, two dogs, and a cat.
          The car and airplane have colors. The first car also has a size.
          No specified negation terms. The background is a garage.
     Objects: [('car', ['white and small', 'yellow']), ('airplane', ['
         yellow']), ('dog', [None, None]), ('cat', [None])]
     Background: A realistic image in a garage
     Negation:

- Example 3
     User prompt: A car and a dog are on top of an airplane and below a
         red chair. There's another dog sitting on the mentioned chair.
     Reasoning: Four objects are described: one car, airplane, two dog,
         and a chair. The chair is red color. No specified negation terms.
          No background is mentioned and thus fill in the default one.
     Objects: [('car', [None]), ('airplane', [None]), ('dog', [None, None
         ]), ('chair', ['red'])]
     Background: A realistic image
     Negation:

- Example 4
     User prompt: An oil painting at the beach of a blue bicycle to the
         left of a bench and to the right of a palm tree with five
         seagulls in the sky.
     Reasoning: Here, there are five seagulls, one blue bicycle, one palm
         tree, and one bench. No specified negation terms. The background
         is an oil painting at the beach.
     Objects: [('bicycle', ['blue']), ('palm tree', [None]), ('seagull', [
         None, None, None, None, None]), ('bench', [None])]
     Background: An oil painting at the beach
     Negation:

- Example 5
     User prompt: An animated-style image of a scene without backpacks.
     Reasoning: The description clearly states no backpacks, so this must
         be acknowledged. The user provides the negative prompt of
         backpacks. The background is an animated-style image.
     Objects: [('backpacks', [None])]
     Background: An animated-style image
     Negation: backpacks

- Example 6
     User Prompt: Make the dog a sleeping dog and remove all shadows in an
          image of a grassland.
     Reasoning: The user prompt specifies a sleeping dog on the image and
         a shadow to be removed. The background is a realistic image of a
         grassland.
     Objects: [('dog', ['sleeping']), ['shadow', [None]]]
     Background: A realistic image of a grassland
     Negation: shadows

- Example 7
     User Prompt: A fire hydrant is back of a cat relative to observer.
         The cat is facing away from the observer.
     Reasoning: Two objects are described: one fire hydrant, and a cat. No
          specified negation terms. No background is mentioned and thus
         fill in the default one.
     Objects: [('fire hydrant', [None]), ['cat', [None]]]
     Background: A realistic image
     Negation: shadows
```

```
Your Current Task: Follow the steps closely and accurately identify
    objects based on the given prompt. Ensure adherence to the above
    output format.
```

Listing 4: Prompt for FoR Interpreter.

```
# Your Role: Expert on spatial relation in multiple perspectives

## Objective: Interpret the prompt and convert the spatial relation into
    the camera's perspective

## Image and Object Specification
1. Image Coordinates: Define square images with top-left at [0, 0] and
    bottom-right at [1, 1].
2. Four of the information objects are given in order, object name,
    bounding box, depth, and facing direction
3. Object Format: (object, box, depth, facing direction)
4. Box Format: [Top-left x, Top-left y, Width, Height]
5. Depth: Define depth of the object from furthest at 0 and nearest at 1.
6. Facing Direction: An orientation of the object relative to the camera
    which can be None, left, forward-left, backward-left, right, forward-
    right, backward-right, front (facing forward or facing toward), or
    back (facing backward of facing away).

## Key Guidelines
1. Perspective Identification: Carefully consider the perspective of the
    spatial relation presented in the prompt.
2. Object facing direction: Carefully consider the facing orientation
    presented in the prompt first, before considering the facing
    orientation from the object specification.
3. Assume the camera, observer, and I (me) are the same thing and have
    the same view (perspective).
4. Look at the example closly to see how the conversion need to make.
<RULES>

## Process Steps
1. Read and understand the user prompt (scene description).
2. Identify the perspective of the spatial relation presented in the
    given prompt.
2. Check whether the facing direction is provided in the prompt.
3. If not, check the facing direction presented in the object
    specification.
4. Explain your understanding (reasoning) and then convert the
    perspective into the camera's perspective
5. If there is no specification of perspective, assume the camera
    perspective for minimal editing of the given prompt.
6. Do not modify other part of the prompt except for spatial relation(s).
7. Do not update the object, only modify the prompt.

## Examples

- Example 1
    User prompt: a backpack on the right of a car from car's perspective
        and a car on the left
    Current Objects: [('backpack #1', [0.302, 0.293, 0.335, 0.194], 0.63,
        None), ('car #1', [0.027, 0.324, 0.246, 0.160]), 0.25, "left"]
    Reasoning: There are two spatial relations presented in the prompt.
        The first one specifies a backpack on the right of a car from "
        the car's perspective." There is no specific the facing diretion
        of the car presented in the prompt. Therefore, consider the car's
         facing direction in the object's current state ("left"). The car
         is facing to the left of the photo. Therefore, the right of the
        car from "car's perspective" is back of the camera. Then, the
        first spatial relation in the camera's perspective is that the
        backpack is back of the car from the camera's perspective. The
```

```
              second spatial relation is a car on the left. This does not
              specify the perspective. Then, assuming a camera perspective for
              this one. Therefore, no update for the second spatial relation.
          Updated prompt: a backpack on the back of a car from camera's
              perspective and a car on the left

      - Example 2
          User prompt: a cat is on the left and the cup is on the right of the
              cat from the cat's view
          Current Objects: [('cat #1', [0.169, 0.563, 0.323, 0.291], 0.901, '
              right'), ('cup #1', [0.59, 0.186, 0.408, 0.814], 0.732, None)]
          Reasoning: There are two spatial relations presented in the prompt.
              The first spatial relation is a cat on the left. The prompt does
              not specify the perspective. Then, assuming a camera perspective
              for this one. Therefore, no update for the first spatial relation
              . The second one specifies the cup is on the right of the cat
              from "the cat's view." There is no specific direction facing the
              cat in the present in the prompt. Therefore, consider the cat's
              facing direction in the object's current state ("right"). The cat
               is facing to the right of the photo. Therefore, the right of the
               cat from "cat's perspective" is front of the camera. Then, the
              second spatial relation in the camera's perspective is that the
              cup on the front of the cat from the camera's view.
          Updated prompt: a cat is on the left and the cup is on the front of
              the cat from the camera's view

      - Example 3
          User prompt: A cow is in front of a sheep from the camera angle. The
              sheep is facing right relative to the camera.
          Current Objects: [('cow #1', [0.354, 0.365, 0.285, 0.385], 0.41, "
              None"), ('sheep #1', [0.608, 0.120, 0.285, 0.200], 0.82, "right")
              ]
          Reasoning: There is only one spatial relation presented in the prompt
              . The prompt specifies that a cow is in front of a sheep from the
               "camera angle." This spatial relation is from the camera's
              perspective. Therefore, there is no need for change.
          Updated prompt: A cow is in front of a sheep from the camera angle.
              The sheep is facing right relative to the camera.

      - Example 4
          User prompt: A fire hydrant is back of a sheep from the sheep's
              perspective. The sheep is facing away from the camera.
          Current Objects: [('fire hydrant #1', [0.113, 0.365, 0.251, 0.251],
              0.64, None), ('sheep #1', [0.608, 0.120, 0.251, 0.251], 0.52, "
              back")]
          Reasoning: There is only one spatial relation presented in the prompt
              . The prompt specifies that a fire hydrant is back of a sheep
              from "the sheep's perspective." The prompt also specifies that
              the sheep is facing away (back) from the camera. So, the back of
              the sheep is the front direction of the camera. The updated
              spatial prompt is a fire hydrant is front of a sheep from the
              camera's perspective.
          Updated prompt: A fire hydrant is front of a sheep from the camera's
              perspective. The sheep is facing away from the camera.

      - Example 5
          User prompt: A deer is to the left of a car from the car's
              perspective. The car is facing away from the camera.
          Current Objects: [('deer #1', [0.454, 0.165, 0.285, 0.385], 0.42,
              None), ('car #1', [0.608, 0.620, 0.285, 0.200], 0.83, "back")]
          Reasoning: There is only one spatial relation presented in the prompt
              . The prompt specifies that a deer is to the left of a car from "
              the car's perspective." The prompt also specifies that the car is
               facing away (back) from the camera. So, the left side of the car
               that is facing away is the left direction of the camera. The
```

```
                     updated spatial prompt is a deer is to the left of a car from the
                        camera's perspective.
                   Updated prompt: A deer is to the left of a car from the camera's
                        perspective. The car is facing away from the camera.

 - Example 6
     User prompt: A cow is to the right of a horse from the horse's
        perspective. The horse is facing toward relative to the camera.
     Current Objects: [('Cow #1', [0.113, 0.365, 0.352, 0.352], 0.83, None
        ), ('horse #1', [0.608, 0.120, 0.352, 0.352], 0.25, "front")]
     Reasoning: There is only one spatial relation presented in the prompt
        . The prompt specifies that a cow is to the right of a horse from
         "the horse's perspective." The prompt also specifies that the
        horse is facing toward (front) the camera. So, the right of the
        horse facing toward is the left direction of the camera. The
        updated spatial prompt is a cow is to the left of a horse from
        the camera's perspective.
     Updated prompt: A cow is to the left of a horse from the camera's
        perspective. The horse is facing toward relative to the camera.

 - Example 7
     User prompt: A deer is in front of a sheep from the sheep's
        perspective. The sheep is facing toward relative to the camera.
     Current Objects: [('deer #1', [0.454, 0.365, 0.285, 0.385], 0.64,
        None), ('sheep #1', [0.608, 0.120, 0.285, 0.200], 0.32, "front")]
     Reasoning:   There is only one spatial relation presented in the
        prompt. The prompt specifies that a deer is in front of a car
        from "the sheep's perspective." The prompt also specifies that
        the sheep is facing toward (front) the camera. So, the front of
        the sheep that faces toward is the front direction of the camera.
         The updated spatial prompt is a deer is in front of a sheep from
         the camera's perspective.
     Updated prompt: A deer is in front of a sheep from the camera's
        perspective. The sheep is facing toward relative to the camera.

 - Example 8
     User prompt: A deer is in front of a dog from the dog's perspective.
        The dog is facing right relative to the camera.
     Current Objects: [('deer #1', [0.186, 0.592, 0.449, 0.408], 0.45, "
        front"), ('dog #1', [0.376, 0.194, 0.624, 0.502], 0.53, "right")]
     Reasoning: There is only one spatial relation presented in the prompt
        . The prompt specifies that a deer is in front of a dog from "the
         dog's perspective." The prompt also specifies that the dog is
        facing to the right of the camera. So, the front of the dog that
        is facing right is the right direction of the camera. The updated
         spatial prompt is a deer is to the right of a dog from the
        camera's perspective.
     Updated prompt: A deer is to the right of a dog from the camera's
        perspective. The dog is facing right relative to the camera.

 - Example 9
     User prompt: A deer is to the right of a car from the car's
        perspective. The car is facing away from the camera.
     Current Objects: [('deer #1', [0.454, 0.165, 0.285, 0.385], 0.42,
        None), ('car #1', [0.608, 0.620, 0.285, 0.200], 0.83, "back")]
     Reasoning: There is only one spatial relation presented in the prompt
        . The prompt specifies that a deer is to the right of a car from
        "the car's perspective." The prompt also specifies that the car
        is facing away (back) from the camera. So, the right side of the
        car that is facing away is the right direction of the camera, don
        't reverse the literal relation like facing toward the camera.
        The updated spatial prompt is that a deer is to the right of a
        car from the camera's perspective.
     Updated prompt: A deer is to the right of a car from the camera's
        perspective. The car is facing away from the camera.
```

```
Your Current Task: Follow the steps closely and accurately convert all
    presented spatial relations in the given prompt into the camera's
    perspective. Ensure adherence to the above output format.
```

Listing 5: Prompt for Layout Interpreter.

```
# Your Role: Expert Bounding Box Adjuster

## Objective: Manipulate bounding boxes in square images according to the
    user prompt while maintaining visual accuracy.

## Object Specifications and Manipulations
1. Image Coordinates: Define square images with top-left at [0, 0] and
    bottom-right at [1, 1].
2. Object Format: (object, box, depth, orientation)
3. Box Format: [Top-left x, Top-left y, Width, Height]
4. Depth: Define depth of the object from furthest at 0 and nearest at 1.
5. Orientation Format: An orientation of the object which can be None,
    Left, Right, Front, or Back.
6. Operations: Include addition, deletion, repositioning, attribute
    modification, and depth modification.

## Key Guidelines
1. Alignment: Follow the user's prompt, keeping the specified object
    count and attributes. Deem it deeming it incorrect if the described
    object lacks specified attributes.
2. Boundary Adherence: Keep bounding box coordinates within [0, 1].
3. Depth Adherence: Keep average depth within [0, 1].
4. Orientation Adherence: An orientation must change depend on the prompt
    . If nothing specify in the prompt, do not change the orientation of
    the object.
5. Minimal Modifications: Change bounding boxes or depth only if they don
    't match the user's prompt (i.e., don't modify matched objects).
6. Overlap Reduction: Minimize intersections in new boxes and remove the
    smallest, least overlapping objects.

## Process Steps
1. Interpret prompts: Read and understand the user's prompt.
2. Implement Changes: Review and adjust current bounding boxes to meet
    user specifications.
3. Explain Adjustments: Justify the reasons behind each alteration and
    ensure every adjustment abides by the key guidelines.
4. Output the Result: Present the reasoning first, followed by the
    updated objects section, which should include a list of bounding
    boxes in Python format.

## Examples

- Example 1
    User prompt: A realistic image of landscape scene depicting a green
        car parking on the left of a blue truck, with a red air balloon
        and a bird in the sky
    Current Objects: [('green car #1', [0.027, 0.365, 0.275, 0.207], 0.6,
        None), ('blue truck #1', [0.350, 0.368, 0.272, 0.208], 0.7, None
        ), ('red air balloon #1', [0.086, 0.010, 0.189, 0.176]), 0.4,
        None]
    Reasoning: To add a bird in the sky as per the prompt, ensuring all
        coordinates and dimensions remain within [0, 1].
    Updated Objects: [('green car #1', [0.027, 0.365, 0.275, 0.207], 0.6,
        None), ('blue truck #1', [0.350, 0.369, 0.272, 0.208], 0.7, None
        ), ('red air balloon #1', [0.086, 0.010, 0.189, 0.176], 0.4, None
        ), ('bird #1', [0.385, 0.054, 0.186, 0.130]), 0.3, None]

- Example 2
```

```
    User prompt: A realistic image of landscape scene depicting a green
        car parking on the right of a blue truck, with a red air balloon
        and a bird in the sky
    Current Output Objects: [('green car #1', [0.027, 0.365, 0.275,
        0.207], 0.79, "left"), ('blue truck #1', [0.350, 0.369, 0.272,
        0.208], 0.68, "right"), ('red air balloon #1', [0.086, 0.010,
        0.189, 0.176]), 0.15, None]
    Reasoning: The relative positions of the green car and blue truck do
        not match the prompt. Swap positions of the green car and blue
        truck to match the prompt, while keeping all coordinates and
        dimensions within [0, 1].
    Updated Objects: [('green car #1', [0.350, 0.369, 0.275, 0.207],
        0.79, "left"), ('blue truck #1', [0.027, 0.365, 0.272, 0.208],
        0.68, "right"), ('red air balloon #1', [0.086, 0.010, 0.189,
        0.176], 0.15, None), ('bird #1', [0.485, 0.054, 0.186, 0.130],
        0.15, "front")]

- Example 3
    User prompt: An oil painting of a pink dolphin jumping on the left of
        a steam boat on the sea
    Current Objects: [('steam boat #1', [0.302, 0.293, 0.335, 0.194],
        0.76, "front"), ('pink dolphin #1', [0.027, 0.324, 0.246, 0.160],
        0.23, "left"), ('blue dolphin #1', [0.158, 0.454, 0.376, 0.290],
        0.26, "right")]
    Reasoning: The prompt mentions only one dolphin, but two are present.
        Thus, remove one dolphin to match the prompt, ensuring all
        coordinates and dimensions stay within [0, 1].
    Updated Objects: [('steam boat #1', [0.302, 0.293, 0.335, 0.194],
        0.76, "front"), ('pink dolphin #1', [0.027, 0.324, 0.246, 0.160],
        0.23, "left")]

- Example 4
    User prompt: An oil painting of a pink dolphin jumping on the left of
        a steam boat on the sea
    Current Objects: [('steam boat #1', [0.302, 0.293, 0.335, 0.194],
        0.76, "front"), ('dolphin #1', [0.027, 0.324, 0.246, 0.160],
        0.23, "left")]
    Reasoning: The prompt specifies a pink dolphin, but there's only a
        generic one. The attribute needs to be changed.
    Updated Objects: [('steam boat #1', [0.302, 0.293, 0.335, 0.194],
        0.76, "front")), ('pink dolphin #1', [0.027, 0.324, 0.246,
        0.160], 0.23, "left")]

- Example 5
    User prompt: a backpack on the right of a car from car's perspective
        and a car on the left
    Current Objects: [('backpack #1', [0.302, 0.293, 0.335, 0.194], 0.63,
        None), ('car #1', [0.027, 0.324, 0.246, 0.160]), 0.25, "left"]
    Reasoning: The prompt specifies that a backpack on the right of "a
        car". There is no specific of orientation of the car from the
        prompt, however, the current car is facing to the left. Therefore
        , the spatial relation from the camera should be that a backpack
        on the back of the car. Average depth of backpack(0.63) is higher
        than a car(0.25) which do not match the prompt. Swap the average
        depth of the car and the backpack to match the prompt, while
        keeping all coordinates and dimensions within [0, 1].
    Updated Objects: [('backpack #1', [0.302, 0.293, 0.335, 0.194],
        0.25, None), ('car #1', [0.027, 0.324, 0.246, 0.160]), 0.63, "
        left"]

- Example 6
    User prompt: a cat is on the left and the cup is on the right of the
        cat from the cat's view
    Current Objects: [('cat #1', [0.169, 0.563, 0.323, 0.291], 0.901, '
        right'), ('cup #1', [0.59, 0.186, 0.408, 0.814], 0.732, None)]
```

```
    Reasoning: The prompt specifies that a cat is on the left, which is
        currently correct. There is no specific of cat's orientation in
        the prompt. Then, the right orientation is acceptable. Then, the
        prompt specififes that a cup is to the right of cat the cat's
        view. This is same as a cup is in front of the cat from camera's
        perspective. However, cup's depth (0.731) is lower than cat's
        depth (0.901). Considering only increasing cup's depth and
        lowering cat's depth, while keeping all coordinates and dimension
         within [0, 1].
    Updated Objects: [('cat #1', [0.169, 0.563, 0.323, 0.291], 0.405, '
        right'), ('cup #1', [0.59, 0.186, 0.408, 0.814], 0.901, None)]

- Example 7
    User prompt: A cow is in front of a sheep from the camera angle. The
        sheep is facing right relative to the camera.
    Current Objects: [('cow #1', [0.354, 0.365, 0.285, 0.385], 0.41, "
        None"), ('sheep #1', [0.608, 0.120, 0.285, 0.200], 0.82, "right")
        ]
    Reasoning: The prompt specifies that a cow is in front of a sheep
        from "the camera angle". Therefore, the spatial relation is that
        a cow is in front of a sheep from the camera's perspective.
        However, the depth of the cow is lower than the sheep, which does
         not match the prompt. Swap the average depth of the cow and the
        sheep to match the prompt, while keeping all coordinates and
        dimensions within [0, 1].
    Updated Objects:  [('cow #1', [0.354, 0.365, 0.285, 0.385], 0.82, "
        None"), ('sheep #1', [0.608, 0.120, 0.285, 0.200], 0.41, "right")
        ]

- Example 8
    User prompt: A fire hydrant is back of a sheep from the sheep's
        perspective. The sheep is facing left relative to the camera.
    Current Objects: [('fire hydrant #1', [0.113, 0.365, 0.251, 0.251],
        0.64, None), ('sheep #1', [0.608, 0.120, 0.251, 0.251], 0.52, "
        left")]
    Reasoning: The prompt specifies that a fire hydrant is back of a
        sheep from "the sheep's perspective". Since the sheep is facing
        to the left of the camera from the prompt, the spatial relation
        from the camera should be that a fire hydrant is right of the
        sheep from the camera's perspective. Therefore, the relative
        positions of the fire hydrant and sheep do not match the prompt
        since the fire hydrant's bounding box is to the left of the sheep
        's bounding box. Swap positions of the fire hydrant and sheep to
        match the prompt, while keeping all coordinates and dimensions
        within [0, 1].
    Updated Objects:[('fire hydrant #1', [0.608, 0.120, 0.251, 0.251],
        0.64, None), ('sheep #1', [0.113, 0.365, 0.251, 0.251], 0.52, "
        left")]

- Example 9
    User prompt: A cow is to the left of a horse from the horse's
        perspective. The horse is facing right relative to the camera.
    Current Objects:  [('Cow #1', [0.113, 0.365, 0.352, 0.352], 0.83,
        None), ('horse #1', [0.608, 0.120, 0.352, 0.352], 0.25, "right")]
    Reasoning: The prompt specifies that a cow is to the left of a horse
        from "the horse's perspective". Since the horse is facing to the
        right of the camera from the prompt, the spatial relation from
        the camera should be that a cow is back of a horse from the
        camera's perspective. However, the depth of the cow (0.83) is
        higher than the horse (0.25), which does not match the prompt.
        Swap the average depth of the cow and the horse to match the
        prompt, while keeping all coordinates and dimensions within [0,
        1].
    Updated Objects: [('Cow #1', [0.113, 0.365, 0.352, 0.352], 0.25, None
        ), ('horse #1', [0.608, 0.120, 0.352, 0.352], 0.83, "right")]
```

```
- Example 10
    User prompt: A deer is in front of a car from the car's perspective.
        The car is facing toward the camera.
    Current Objects: [('deer #1', [0.454, 0.365, 0.285, 0.385], 0.64,
        None), ('car #1', [0.608, 0.120, 0.285, 0.200], 0.32, "left")]
    Reasoning: The prompt specifies that a deer is in front of a car from
         "the car's perspective". Since the car is facing toward the
        camera from the prompt, the spatial relation from the camera
        should be that a deer is in front of a car from the camera's
        perspective. Average depth of deer (0.64) is higher than average
        depth of cow (0.32), match the prompt. However, the orientation
        of the car is left. The orientation of car need to be changed.
    Updated Objects: [('deer #1', [0.454, 0.365, 0.285, 0.385], 0.64,
        None), ('car #1', [0.608, 0.120, 0.285, 0.200], 0.32, "front")]

- Example 11
    User prompt: A deer is in front of a car from the car's perspective.
        The car is facing away from the camera.
    Current Objects: [('deer #1', [0.454, 0.165, 0.285, 0.385], 0.42,
        None), ('car #1', [0.608, 0.620, 0.285, 0.200], 0.83, "back")]
    Reasoning: The prompt specifies that a deer is in front of a car from
         "the car's perspective". Since the car is facing away from the
        camera from the prompt, the spatial relation from the camera
        should be that a deer is back of a car from the camera's
        perspective. Average depth of deer is lower than average depth of
         cow. Thus, the image aligns with the user's prompt, requiring no
        further modifications.
    Updated Objects: [('deer #1', [0.454, 0.165, 0.285, 0.385], 0.42,
        None), ('car #1', [0.608, 0.620, 0.285, 0.200], 0.83, "back")]

- Example 12
    User prompt: A realistic photo of a scene with a brown bowl on the
        right and a gray dog on the left
    Current Objects: [('gray dog #1', [0.186, 0.592, 0.449, 0.408], 0.45,
         "front"), ('brown bowl #1', [0.376, 0.194, 0.624, 0.502], 0.53,
        None)]
    Reasoning: The leftmost coordinate (0.186) of the gray dog's bounding
         box is positioned to the left of the leftmost coordinate (0.376)
         of the brown bowl, while the rightmost coordinate (0.186 +
        0.449) of the bounding box has not extended beyond the rightmost
        coordinate of the bowl. Thus, the image aligns with the user's
        prompt, requiring no further modifications.
    Updated Objects: [('gray dog #1', [0.186, 0.592, 0.449, 0.408], 0.45,
         "front"), ('brown bowl #1', [0.376, 0.194, 0.624, 0.502], 0.53,
        None)]

Your Current Task: Carefully follow the provided guidelines and steps to
    adjust bounding boxes in accordance with the user's prompt. Ensure
    adherence to the above output format.
```

Listing 6: Prompt for Extract Cognative Map before Generating Image.

```
Please also consider performing the cognitive map before generating the
    image.

Cognative Map's objective is to identify specific objects from the
    context,
understand the spatial arrangement of the scene, and estimate the center
    point of each object, assuming the entire scene is represented by a
    10x10 grid.

[Rule]
1. Estimate the center location of each instance within the provided
    categories, assuming the entire scene is represented by a 10x10 grid.
2. If a category contains multiple instances, include all of them.
```

```
3. Each  o b j e c t s  estimated location should accurately reflect its real
   position in the scene, preserving the relative spatial relationships
   among all objects.

Still, you need to generate the real image, not a cognitive map.
```

