# OpenReview forum: "FoR-SALE: Frame of Reference-guided Spatial Adjustment in LLM-based Diffusion Editing"
_ICLR.cc/2026/Conference — Submitted to ICLR 2026_

### Official Review · Reviewer_Y3Ka · 2025-10-18

**Soundness:** 3
**Presentation:** 3
**Contribution:** 2
**Rating:** 4
**Confidence:** 4

**Summary:**

This paper addresses a critical limitation in Text-to-Image (T2I) generation: the inability of current methods to correctly render spatial relationships described from perspectives other than the default camera view. The authors propose FoR-SALE, a self-correction framework, including visual perception modules for spatial analysis, LLM-based reasoning for perspective interpretation and layout planning, and diffusion-based operations for iterative refinement.

**Strengths:**

*   The investigation of spatial relations from diverse perspectives is compelling. The proposed vision perception modules and LLM-based interpretation seem reasonable, as they provide 3D prior information.
*   The experimental results show the method's efficacy across multiple T2I baselines. Furthermore, the proposed progressive refinement strategy can further enhance the generated spatial accuracy.

**Weaknesses:**

*   The core components of the self-correction strategy and the LLM-based parser appear to build directly upon established paradigms [1], which may limit the methodological novelty. Furthermore, could current Spatial-LLMs [2,3] be potentially applied for enhanced spatial reasoning in this framework? If so, more discussion and comparison with these approaches should be included.

*   It remains unclear whether the proposed method can effectively handle complex scenes involving multiple objects with intricate relations or cases where objects overlap. If such scenarios significantly impact the method's performance, this should be acknowledged and discussed as part of the limitations.

*   The computational efficiency of the approach requires further clarification. What is the specific time overhead of the proposed framework? Does multi-round refinement lead to substantially increased inference time?

References

[1] L. Lian et al., "LLM-grounded Diffusion: Enhancing Prompt Understanding of Text-to-Image Diffusion Models with Large Language Models," arXiv, 2023.

[2] J. Yang et al., "Thinking in Space: How Multimodal Large Language Models See, Remember, and Recall Spaces,"  CVPR, 2025.

[3] Y. Mao et al., "SpatialLM: Training Large Language Models for Structured Indoor Modeling," arXiv, 2025.

**Questions:**

Please address my concerns in the **Weaknesses** section. I would change my rating if my concerns could be addressed.

---

> ### Author Response · Authors · 2025-11-22
>
> We appreciate your thoughtful feedback and valuable insights, and for recognizing our studies. Below, we respond to weaknesses, comments, and questions.
>
> ---
>
> # [W1] Contribution of Framework and Impact of Current Spatial-LLM
>
> Our idea is to solve the FoR problem at the reasoning level rather than at the perception level. We believe this problem is even harder to solve at the low-perception level, as Table 1 shows that even GPT-4o struggles with it. Current models do not have access to the appropriate data with embodied interaction to address this issue at large scale. To rephrase, our contribution is to address the lack of spatial Frame of Reference understanding at a higher level through reasoning (i.e., without training) while leveraging existing lower-level vision tools. This is why we naturally use established paradigms and extend them to cover this novel dimension (Frame of Reference), which heavily impacts spatial reasoning in realistic and interactive environments.
>
> Regarding the comment about using current Spatial-LLMs to enhance the spatial reasoning of the proposed framework, we have conducted experiments with o3 and o4-mini, two recent SOTA models that claim to have improved ability to interpret spatial relationships due to their targeted training. The results in Table 2 indicate that although these models achieve significantly higher performance in relative Frame of Reference (camera perspective) contexts, their performance is still very low when reasoning over the intrinsic Frame of Reference. More analysis is provided in Research Question 3 (RQ3).
> We believe this behavior is expected for current LLMs that claim spatial understanding, due to their nature of training data (perhaps not exposed to interactive training in embodied situations) and their capabilities remain limited to the relative FoR.
>
> Regarding the two suggested frameworks:
>
> Yang J. [2] proposes prompt engineering to extract cognitive maps for spatial reasoning over video. This is not directly related to our framework. Nonetheless, we apply this prompting approach as an additional baseline using GPT-4o to examine whether the cognitive map contributes to improvements in image generation respecting the FoR specifications. We will update the results based on this setting.  We expect the model to still have the problem regarding FoR, which requires 3D understanding (depth and perspective), since the cognitive map is only based on the 2D representation.
>
> For SpatialLM [3], the model is extensively trained to perform 3D object detection and reconstruct 3D structure using point cloud representation as input; it does not apply to our setting. Our setting focuses on interpreting linguistic expressions of FoR in image generation (we don't have any initial image input).
>
> ---
>
> # [W2] Handling Complex Scene
>
> We completely agree that our dataset and the type of scenes are simple -looking into this problem for the first time. However, the results of both the baseline and our method show that current SOTA models (e.g., FLUX.1, GPT-4o) exhibit shortcomings in understanding spatial relations, especially for non-camera perspectives (intrinsic). Correcting these simple cases remains challenging, and handling complex scenes with multiple objects is an open direction for future work. We will include a statement regarding simple and complex scenes in the limitations section.
>
> # [W3] Computation Efficiency
>
> Please see the general comment regarding execution costs to address this weakness (GR1).
>
> ---
>
> We will add some of this information to the main content and to the appendix in the updated version of the paper. Thank you once again for acknowledging our work. **We would be grateful if you could increase some of your scores if our responses address your concerns.**

---

> > ### Author Response · Authors · 2025-11-23
> >
> > # Additional Results with Cognitive Map
> >
> > | Method          | FoR-LMD Relative | FoR-LMD Intrinsic | FoR-LMD Avg | FoREST Relative | FoREST Intrinsic | FoREST Avg. | Avg.
> > | --- | --- | --- | ---- | ---- | ---- | ---- | ---- |
> > |GPT-4o                                 | 94.76 | 24.35 | 56.60 | 57.81 | 35.56 | 47.80 |  52.20
> > | + Cognitive Map                 | 98.23 | 30.36 | 62.00 | 56.00 | 37.33 | 47.60 |  54.80
> > | + 1 round FoR-SALE (Our) | 93.01 | 35.42 | 61.80 | 54.18 | 37.33 | 46.60 |  54.20
> >
> > These are the results regarding the use of prompt engineering to extract a cognitive map before generating the image with GPT-4o compared to 1 round of applying FoR-SALE. We observe that the model achieves a significant gain on FoR-LMD, especially on the relative FoR.
> > This is expected, as the cognitive map should improve 2D spatial relations. We further compared the errors in FoR-LMD based on the output of our automatic evaluator to investigate the source of this improvement. We found that the cognitive map helps the model predict a more accurate number of objects, which contributes to the improvement of FoR-LMD in both the relative and intrinsic FoR. However, this model with the cognitive map still struggles with FoR reasoning, leading to more errors in spatial relations. This is where our proposed framework helps the most.
> >
> > |Error Type         | Cognitive Map | FoR-SALE |
> > | ---- | ----- | ----- |
> > |Missing Obj        |    7             |       10  |
> > |Multiple Obj       |    38           |      75   |
> > |Spatial Relation  |    173         |    144   |
> >
> >
> > **We are adding this information in the updated version of paper.**

---

### Official Review · Reviewer_gDU6 · 2025-10-28

**Soundness:** 3
**Presentation:** 3
**Contribution:** 3
**Rating:** 6
**Confidence:** 3

**Summary:**

This paper introduces FoR-SALE, a framework for improving spatial consistency in text-to-image generation when the textual description uses a non-camera frame of reference (FoR). Built upon the SLD pipeline, FoR-SALE integrates a rule-based FoR-Interpreter for linguistic normalization, orientation, and depth estimation for visual grounding, and two latent editing modules for iterative correction. Experiments on two small benchmarks (FoR-LMD and FoREST) show consistent but moderate performance improvements.

**Strengths:**

The paper targets a clear and practically relevant problem that has received little prior attention: spatial reasoning under varying frames of reference.
1. The paper frames a well-motivated problem clearly and proposes a reasonable solution path.

2. The pipeline demonstrates robust engineering design, combining rule-based linguistic grounding with structured latent-space edits.

3. The presentation is clear, figures are intuitive, and the authors provide sufficient implementation details to facilitate reproducibility.

**Weaknesses:**

1. This method is largely an integration of existing components, such as SLD, SAM, ControlNet, and depth/orientation estimators.

2. The reported gains, though consistent, are relatively modest and require multiple editing rounds.

3. The FoR-Interpreter relies on 32 handcrafted rules, limiting generalization to complex or ambiguous spatial relations, and the benchmarks used are small and partly synthetic.

4. The ablation results suggest that genuine 3D structural correction is still only partially achieved.

**Questions:**

1. How can the 32-rule FoR-Interpreter scale to more complex or compositional spatial relations?

2. Have you considered replacing the rule-based mapping with a learnable FoR normalization module for better generalization?

3. Please include a detailed efficiency analysis to clarify the computational cost and practicality of the proposed iterative pipeline.

4. See in W4.

**Details Of Ethics Concerns:**

No.

---

> ### Author Response · Authors · 2025-11-22
>
> We appreciate your thoughtful feedback and valuable insights, and for recognizing our studies. Below, we respond to weaknesses, comments, and suggestions.
>
> ---
>
> # [W1 & W2] Limitation of Novelty and Performance Gain
>
> We are thankful that the reviewer recognized our contribution in addressing an important problem that has not received enough attention.
> Our approach tackles this problem at the level of reasoning rather than at the perception level.
> Since current models have not been trained with the appropriate data to recognize the FoR, we build on the capabilities of existing models and propose a method that reasons over the frame of reference and edits the image accordingly.
> Of course, as low-level perception tools improve, our framework will be able to reason over more accurate information and improve as well.
> Furthermore, we believe that higher-level Frame of Reference reasoning is harder to address at the perception level—we see that even GPT-4o struggles with this problem in Table 1—compared to low-level tasks such as object detection and orientation estimation.
> This highlights the benefit of our framework, which integrates low-level vision analysis to perform higher-level reasoning for FoR editing.
>
> ---
>
> # [W3 / Q1] Generalizability of Rules in Complex Spatial Relation
>
> Please refer to the general comment about the generalizability of rules in complex spatial relations to address this weakness [GR2].
>
> ---
>
> # [W4] Limitation of 3D editing
>
> Editing depth and facing direction (3D structure) remains challenging in diffusion-based methods. Recent papers have specifically proposed methods to address only the 3D awareness issue [1, 2] in the initial generation, which can still contain errors.
> Our goal is to provide a framework that can reason about Frames of Reference expressed in natural language and map them to depth/orientation correction operations by comparing them to a generated image and verifying the errors. Our depth-editing and orientation-edit operations themselves show effectiveness, though they can be improved.  We believe this framework remains a significant contribution of our proposed method.
>
> [1] Parihar R., et al. Compass Control: Multi-Object Orientation Control for Text-to-Image Generation. CVPR 2025
>
> [2] Le D., et al. One Diffusion to Generate Them All. CVPR 2025
>
> ---
>
> # [Q2] Learnable module for frame of reference (FoR) rules
>
> Replacing the FoR interpreter with a learnable module is a promising future direction. However, such a module would require large-scale, supervised examples of FoR reasoning.
> Current benchmarks that focus directly on FoR consider only 2–3 objects with limited spatial relations [1, 2, 3]. These datasets are insufficient to train a generalizable FoR normalization model, especially in complex scenarios. A more comprehensive dataset focused specifically on FoR reasoning would be needed before such a learnable module can be meaningfully developed and compared. This is a very interesting future direction, particularly for collecting training data via embodied agents in 3D environments.
>
> [1] Zhang et al. Do vision-language models represent space and how? evaluating the spatial frame of reference under ambiguities, 2025
>
> [2] Premsri et al. FoREST: Frame of reference evaluation in spatial reasoning tasks, 2025
>
> [3] Wang et al. Genspace: Benchmarking spatially-aware image generation, 2025
>
> ---
>
> # [Q3] Exeuction Time Analysis
>
> Please see the general comment regarding execution costs to address this weakness (GR1).
>
> ---
>
> ***We will add some of this information to the main content and to the appendix in the updated version of the paper. Thank you once again for acknowledging our work. We would be grateful if you could increase some of your scores if our responses address your concerns.***

---

### Official Review · Reviewer_XXL1 · 2025-10-28

**Soundness:** 4
**Presentation:** 4
**Contribution:** 3
**Rating:** 6
**Confidence:** 4

**Summary:**

This paper tackles a significant limitation in modern Text-to-Image (T2I) models: their poor understanding of spatial relations described from a non-camera (intrinsic) Frame of Reference (FoR). While models can often place objects correctly relative to the camera (e.g., "a horse to the left of a cow"), they fail when the perspective is object-centric (e.g., "a horse to the left of a cow *from the cow's perspective*").

To address this, the authors propose **FoR-SALE** (Frame of Reference-guided Spatial Adjustment in LLM-based Diffusion Editing), a framework that extends the Self-correcting LLM-controlled Diffusion (SLD) pipeline.

**Strengths:**

- The paper clearly identifies and addresses the failure of SOTA T2I models on non-camera (intrinsic) Frames of Reference, a key component of spatial reasoning.
- The FoR-SALE framework's design is logical. The **FoR Interpreter** is a particularly strong contribution, as it cleverly simplifies the reasoning problem by translating intrinsic prompts into a canonical camera-centric view *before* layout planning.
- The paper introduces `Facing Direction Modification` and `Depth Modification`, two new latent-space operations that are necessary for correcting the 3D spatial errors (orientation, depth) inherent in FoR tasks.

**Weaknesses:**

- The framework's performance is heavily capped by the quality of its sub-modules, which the paper acknowledges. The failure analysis (Appendix F) shows that "Incorrect orientation generation" (28.33%) and "Object Detection failure" (18.33%) are the dominant error sources. The framework can't fix what it can't "see" or what the underlying diffusion model can't "draw" correctly.
- The ablation study (RQ4, Table 3) shows that removing the `Depth Modification` operation results in only a minor 0.3% accuracy drop. This suggests the current implementation (using ControlNet) has a very limited positive impact, and correcting 3D depth remains an unsolved challenge.
- The FoR-Interpreter relies on 32 manually-defined "perspective conversion rules" (Appendix G). This approach, while effective for the 4 relations (L/R/F/B) and 8 directions tested, seems brittle. It's unclear how this would scale to more complex spatial prepositions (e.g., "between," "alongside," "diagonally behind") or more nuanced descriptions without a combinatorial explosion of rules.
- The evaluation is entirely automated. While the authors state the protocol is based on prior work aligned with human judgment, the complexity of spatial relations (especially FoR) can be subtle. A small-scale human study to validate that the automated "correct" images are indeed perceptually correct would strengthen the paper's claims.

**Questions:**

1. The ablation study (RQ4) shows that `Depth Modification` has a minimal (0.3%) impact on performance. Why is this contribution so small? Is it a limitation of the underlying ControlNet-based operation, or do you believe depth is simply less critical than orientation for the spatial relations in your benchmarks?
2. he FoR-Interpreter's 32 manual rules are a core component. How do you see this approach scaling to more complex spatial language, such as relations involving three or more objects (e.g., "A is between B and C") or more nuanced prepositions ("A is just to the left of B")? Does this manual-rule approach present a fundamental bottleneck?
3. Your failure analysis (Appendix F) identifies "Incorrect orientation generation" (28.33%) as the single largest source of error after one round of correction. Since your `Facing Direction Modification` operation directly targets this, could you provide more insight into why this operation fails so often? Is the issue with the DiffEdit model used, or is it a problem of composing the new latent representation back into the scene?

---

> ### Author Response · Authors · 2025-11-22
>
> We appreciate your positive feedback and valuable insights. Below, we respond to weaknesses, comments, and suggestions.
>
> ---
>
> # [W1] Limitation of Framework by Sub-Modules Performance
>
> This is indeed correct—the performance of the proposed framework is bounded by the quality of its sub-modules.
> Our idea is to solve the frame-of-reference comprehension problem by leveraging high-level reasoning and correction in image generation. We believe this problem is even more difficult to solve at the low-perception level—Table 1 shows that even GPT-4o struggles with this issue. A possible future direction is to collect large amounts of data with 3D interactions (which is challenging in itself) and train models to handle the FoR problem at the perception level. To rephrase, our contribution is to address the lack of spatial frame-of-reference understanding at a higher level through reasoning while leveraging existing lower-level vision tools. Certainly, our framework will continue to improve as the underlying tools improve.
>
> ---
>
> # [W2/Q1] Limitation of Depth Editing
>
> A minor 0.3% accuracy drop highlights the gain from the first round of editing only. The difference could potentially be higher when we use more rounds of editing with FoR-SALE. We provide the results of applying FoR-SALE without the Depth Modification operation in the second round, where the accuracy drops by 2% overall.
> This shows that the Depth Modification still has an impact during further refinement, even if it appears to have less influence in the first round.
>
> | Method     | Relative | Intrinsic | Average |
> | ----------- | ----------| ----------| --------- |
> |FLUX.1                                      |38.57     | 20.70     | 29.00 |
> |FLUX1 + FoR-SALE (2 Round)                  |47.22     | 28.83     | 38.10 |
> |FLUX1 + FoR-SALE w/o Depth Edit (2 Round)   |45.83     | 26.21     | 36.10 |
>
> Editing depth and facing direction (3D structure) remains challenging in diffusion-based methods. Recent papers have specifically proposed methods to address only the 3D awareness issue [1, 2] in the initial generation, which can still contain errors.
> Our goal is to provide a framework that can reason about Frames of Reference expressed in natural language,  and map it to depth/orientation correction operations by comparing to a generated image and verifying the errors. Our depth-editing operation itself shows effectiveness, though it can be improved.  We believe this framework remains a significant contribution of our proposed method.
>
> [1] Parihar R., et al. Compass Control: Multi-Object Orientation Control for Text-to-Image Generation. CVPR 2025
> [2] Le D., et al. One Diffusion to Generate Them All. CVPR 2025
>
> ---
>
> # [W3/Q2] Generalizability of Rules in Complex Spatial Relation
>
> Please refer to the general comment about the generalizability of rules in complex spatial relations to address this weakness [GR2].
>
> ---
>
> # [W4] Human Alignment of the Automatic Evaluator
>
> We perform a small study of human evaluation compared to automatic evaluation over 200 samples measuring the total correctness of the generated image.
> We calculate the Cohen-Kappa alignment score, which measures agreement between two evaluators (automatic and human). The results show that the Cohen's kappa is 0.791, which is acceptable given the agreement between human judgment and our automatic evaluation. **We will add this information to the paper.** The accuracy reported in the human evaluation (32\%) is much higher than that reported by automatic calculation (28\%) for the selected subset. This means our model performs slightly better than the numbers reflected in the automatic evaluation metric reported in our tables.
>
> ---
>
>  # [Q4] Failure of Orientation Generation
>
> To provide insight into DiffEdit’s orientation editing, we examined the images generated by DiffEdit in 1 round of FoR-SALE across three baselines that solely edited the orientation of a single object.
> We report the error percentage of DiffEdit orientation edits below. The soft-error metric allows more orientation labels to be considered correct (e.g., treating “left” as correct for backward-left and forward-left, and vice versa), whereas the hard-error metric requires the generated orientation to match the expected label exactly.
>
> | Baseline | Soft-Error |  Hard Error. |
> | ------ | ----- | ----- |
> | SD3.5      | 56.25      |      73.20  |
> | FLUX.1     | 67.1       |      78.71  |
> | GPT4o      | 49.47      |      72.35  |
>
> These results show the proportion of “Incorrect orientation generation” errors that stem from DiffEdit itself. They indicate that most of the errors arise because DiffEdit cannot generate the object with the correct orientation, rather than from failures in composing the latent space.
>
> ---
>
> ***We will add this information to the updated version. Thank you once again for acknowledging our work. We would be grateful if you could increase some of your scores if our responses address your concerns.***

---

> > ### Comment · Reviewer_XXL1 · 2025-11-27
> >
> > I appreciate the authors' rebuttal, and I will maintain my original positive evaluation. Thank you.

---

### Official Review · Reviewer_Tqds · 2025-11-01

**Soundness:** 3
**Presentation:** 3
**Contribution:** 2
**Rating:** 4
**Confidence:** 3

**Summary:**

This paper focuses on the reasoning deficiencies of multimodal large models (MLLMs) in handling Frames of Reference (FoR), particularly in text-to-image (T2I) generation, where spatial misalignment frequently occurs when the described scene adopts a non-camera viewpoint. To address this issue, the authors propose FoR-SALE, a self-correction approach that explicitly models FoR within an LLM-guided diffusion editing framework. The method performs edits in the latent space of diffusion models and supports multi-round iterative refinement. The authors evaluate their approach on the FoR-LMD and FoREST benchmarks, achieving state-of-the-art performance.

**Strengths:**

1. The FoR plays a crucial role in spatial reasoning of MLLMs, yet it has not been widely explored in prior research.
2.The authors conducted various analyses in the experimental section, such as error analysis,which enhances the rigor of the paper.

**Weaknesses:**

1. In Table 1, the performance of GraPE and SLD declines compared to the vanilla model. Does this indicate that these compared methods are relatively weak?
 2. In Line 313, The authors seem to have reused the image understanding component of their method in the evaluation criteria. Could this introduce an evaluation bias and ultimately lead to higher scores for their proposed approach?
 3. Although I am very satisfied with the series of analyses conducted by the authors, is there a lack of additional generation latency metrics to assess whether the improvements are worthwhile? After all, image generation time is also an important factor to consider.

**Questions:**

See weakness.

---

> ### Author Response · Authors · 2025-11-21
>
> We appreciate your thoughtful feedback and valuable insights, and for recognizing our studies. Below, we respond to weaknesses, comments, and suggestions.
>
> ---
>
> ## [W1] Low Performance of GraPE and SLD
>
> The lower performance of GraPE and SLD in Table 1 occurs when the input context uses the Intrinsic Frame of Reference (FoR), where spatial relations are described from the reference object’s perspective.
> In contexts that use the relative FoR (camera's perspective), these two baselines still show better performance compared to the vanilla models (SD3.5 and FLUX.1).
> So, this indicates that these methods are not weak in general, but they are designed specifically to edit spatial expressions tied to the camera perspective.  This result reveals the lack of FoR comprehension from various perspectives, which is the main contribution of our proposed method. Table 1 also shows that our FoR-SALE provides improvement over the vanilla models on the intrinsic FoR.
>
> ---
>
> ## [W2] Evaluation Bias
>
> We agree that changing the evaluation modules to different stacks may change the accuracy. However, as explained in Section 4.4, we incorporate a different stack of evaluation modules in Appendix D.  The accuracy increased by 2% when using a different evaluation stack.
> FoR-SALE outperforms consistently even when changing the evaluation modules (4\% over the initial images from FLUX.1). This confirms that our evaluation is robust and reliable.
>
> ---
>
> ## [W3] Execution Time Analysis
>
> Please see the general comment regarding execution costs to address this weakness (GR1).
>
> ---
>
> ***We will add some of this information to the main content and to the appendix in the updated version of the paper. Thank you once again for acknowledging our work. We would be grateful if you could increase some of your scores if our responses address your concerns.***

---

### Author Response · Authors · 2025-11-21
**General Response**

Thank you to all the reviewers for your thorough comments and insights. We would like to address some of the commonly raised issues in this general response. **We will provide specific responses to each weakness and question in separate comments for each reviewer.**

---

# GR1: Execution Costs

We provide the execution time for each component in the table below, averaged over 1 round of applying FoR-SALE to the initial image from FLUX.1. We also provide the time estimate for generating an image from FLUX.1 as the baseline.
All times are recorded on 1 GPU A6000 with 48GB.


|  Component     | Time (s) |
| :------- | :------: |
|   FLUX.1              |   49.52  |
| LLM Parser            | 21.64    |
| Object Detection      | 0.94     |
| Depth Detection       | 1.23     |
| Orientation Detection | 1.60     |
| FoR Interpreter       | 31.84    |
| Layout Interpreter    | 45.52    |
| Latent Operations     | 17.91    |
| Generate new image    | 7.99     |
| FoR-SALE  (Accumulate all components)            | 128.67   |


According to our time efficiency analysis, we observe up to 2.5× higher time consumption compared to the baseline image generation (FLUX.1). Three components contribute most to the overhead, all of which are LLM-based.
The LLM Parser used to extract objects from the sentence will be called only in the first round, which would save a lot of time for the multi-round editing. The Layout Interpreter is the most time-consuming component, as the LLM must process and propose the layout. We believe this cost is comparable to the original SLD framework, which also includes these two components (i.e., Parser and interpreter).
The new component introduced by our method, that is the FoR Interpreter, increases the time consumption by only around 0.5× relative to the baseline we built upon. This trade-off is a reasonable choice for enabling the framework to understand spatial relations from non-camera perspectives—a challenging aspect that current methods largely overlook—and achieve a better accuracy (5\% when applying 1 round of FoR-SALE) compared to SLD under the evaluation benchmarks.

Other components consume relatively small amounts of time compared to these three LLM modules. **We are going to update this report in the paper to provide the execution cost of our proposed framework.**

----

# GR2: The generalizability of rules in complex spatial scenarios


The more complex spatial relations raised by the reviewer (e.g., between, overlap, alongside) are not directly influenced by the Frame of Reference. Our 32 manually defined rules are designed specifically for the basic spatial relations that are affected by FoR changes, as described in Section 3.2.1. We also conducted an ablation study on partial rule usage and found that these rules—even when not complete—help the LLM perform perspective conversion correctly, as demonstrated by the significant improvement shown in Table 2 (compared to using no rules). **Given that more complex relations often build on the same left/right/front/back transformations, we believe our rules provide a foundation for perspective conversion even in more complex cases.**

---

### Author Response · Authors · 2025-12-04
**Summary of Strength and Revision of the paper**

We appreciate the reviewers’ time and insightful feedback on our paper. Below, we summarize the key strengths highlighted across the reviews and the revisions made in the updated version of the paper. For detailed, point-by-point responses, please refer to the individual reviewer replies.

### Strengths:

1. **Identification of a critical and underexplored problem** — The paper clearly identifies and addresses failures in Frame-of-Reference (FoR) reasoning, an important yet insufficiently studied limitation in current MLLMs. (All reviewers)

2. **Comprehensive experimental analysis** — The paper provides extensive analysis that strengthens the rigor and clarity of the experimental section. (Reviewer Tqds)

3. **Logical and effective framework design** — The FoR-SALE framework is well-motivated, and its FoR Interpreter is recognized as a key contribution for improving FoR-based spatial understanding. (Reviewers XXL1, gDU6)

4. **Introduction of necessary 3D editing components** — The inclusion of Facing Modification and Depth Modification, adapted from existing approaches, is viewed as essential for correcting 3D spatial errors. (Reviewer XXL1)

5. **Clear presentation and reproducibility** — The paper is well-written, clearly organized, and includes sufficient details for reproduction. (Reviewer gDU6)

6. **Consistent improvements across baselines** — The framework demonstrates performance improvements over multiple baselines. (Reviewer Y3Ka)

---

### Revisions to the Paper (highlighted in blue color in the revised version):

1. **Execution Cost** — To address concerns regarding the time efficiency of our framework, we included additional analysis in Section 5 that reports the execution time of each component of FoR-SALE.

2. **Limitation of Depth Edit** — We added new results in Table 3 comparing two rounds of FoR-SALE without DepthEdit and expanded the corresponding discussion in RQ4. This directly addresses reviewer XXL1’s comment that DepthEdit may contribute minimally in the first round. Our updated analysis shows that this component has a more substantial impact in subsequent rounds, further confirming its effectiveness within the full framework.

3. **Human Alignment Score** — We added details of a small-scale human study in Appendix H, with corresponding results summarized in Section 4.2. These findings demonstrate strong alignment between human judgments and our automatic evaluation, confirming its validity.

4. **Failure of Orientation Generation** — We expanded our failure analysis in Appendix F by quantifying orientation errors using both soft and hard metrics.

5. **Additional Baselines Using Current Spatial-LLM Methods**— We incorporated a prompt engineering baseline based on the cognitive map extraction proposed in [1] (Table 1). We discussed this result in Section 5 and provided further discussion in Appendix C.3.

The remaining clarifications address reviewers’ questions about motivations and methodological choices, drawing on information already in the paper. Detailed responses to each point are provided in the individual reviewer replies.


[1] J. Yang et al., "Thinking in Space: How Multimodal Large Language Models See, Remember, and Recall Spaces," CVPR, 2025.

---

### Meta-Review · Area_Chair_25sr · 2025-12-16

**Summary:**

The paper aims at addressing the limitation in modern T2I models when spatial descriptions are provided from non-camera (intrinsic) Frame of Reference (FoR) by proposing FoR-SALE that extends the self-correcting LLM-controlled Diffusion pipeline.

Strengths identified by reviewers include, its crucial and well-motivated research problem, strong contribution of FoR Interpreter, two new latent-space operations and clear presentation.

The reviewers raise several concerns and the major concern is that the FoR-Interpreter relies on 32 manually-defined rules and this limits its generalization ability to more complex spatial relations. Additionally, the ablation study indicates that the positive impact of control net is  limited and correcting 3D depth remained unsolved. One reviewer also pointed out that the benchmarks used in the paper are small and partly synthetic.

Given these considerations, this paper fails to meet the standards of ICLR. The authors are encouraged to address the highlighted issues to strengthen their contribution to the field of FoR.

**Reviewer Concerns:**

The authors have provided feedback and additional results during the rebuttal and have revised the manuscript accordingly. The feedback has addressed most concerns including, 1) weak compared methods (GraPE and SLD have lower performance  compared to the vanilla model); 2) Evaluation bias as the reuse of image understanding component of the proposed method; 3) Time cost analysis; 4) framework's performance is heavily capped by the quality of its sub-modules; 5) Failure of Orientation Generation; 6) limited novelty as the method is an integration of existing techniques; 7) effect of integrating current Spatial-LLM.

However, there are concerns that are not fully addressed: a) limited positive impact of control net and unsolved challenge of correcting 3D depth; b) generalization to more complex spatial relations as FoR-Interpreter relies on 32 manually-defined rules; c) benchmarks used are small and partly synthetic.

**Reviewer Scores:**

Initially, the reviewer scores are mixed with two borderline accepts and two borderline rejects. Reviewer XXL1 has confirmed maintaining initial score and other reviewers would have maintained their scores as some concerns have not been addressed.

---

### Decision · Program_Chairs · 2026-01-26

Reject